# Control of neurotransmitter release by two distinct membrane-binding faces of the Munc13-1 $C_1C_2B$ region

Marcial Camacho[1,2†], Bradley Quade[3,4,5†], Thorsten Trimbuch[1,2], Junjie Xu[3,4,5], Levent Sari[3,6,7], Josep Rizo[3,4,5]*, Christian Rosenmund[1,2]*

[1]Charité – Universitätsmedizin Berlin, corporate member of Freie Universität Berlin and Humboldt-Universität zu Berlin, Institute of Neurophysiology, Berlin, Germany; [2]NeuroCure Cluster of Excellence, Berlin, Germany; [3]Department of Biophysics, University of Texas Southwestern Medical Center, Dallas, United States; [4]Department of Biochemistry, University of Texas Southwestern Medical Center, Dallas, United States; [5]Department of Pharmacology, University of Texas Southwestern Medical Center, Dallas, United States; [6]Cecil H. and Ida Green Comprehensive Center for Molecular, Computational and Systems Biology, University of Texas Southwestern Medical Center, Dallas, United States; [7]Center for Alzheimer's and Neurodegenerative Diseases, University of Texas Southwestern Medical Center, Dallas, United States

**\*For correspondence:**
Jose.Rizo-Rey@UTSouthwestern.edu (JR);
christian.rosenmund@charite.de (CR)

†These authors contributed equally to this work

**Competing interest:** The authors declare that no competing interests exist.

**Abstract** Munc13-1 plays a central role in neurotransmitter release through its conserved C-terminal region, which includes a diacyglycerol (DAG)-binding $C_1$ domain, a $Ca^{2+}$/$PIP_2$-binding $C_2B$ domain, a MUN domain and a $C_2C$ domain. Munc13-1 was proposed to bridge synaptic vesicles to the plasma membrane through distinct interactions of the $C_1C_2B$ region with the plasma membrane: (i) one involving a polybasic face that is expected to yield a perpendicular orientation of Munc13-1 and hinder release; and (ii) another involving the DAG-$Ca^{2+}$-$PIP_2$-binding face that is predicted to result in a slanted orientation and facilitate release. Here, we have tested this model and investigated the role of the $C_1C_2B$ region in neurotransmitter release. We find that K603E or R769E point mutations in the polybasic face severely impair $Ca^{2+}$-independent liposome bridging and fusion in in vitro reconstitution assays, and synaptic vesicle priming in primary murine hippocampal cultures. A K720E mutation in the polybasic face and a K706E mutation in the $C_2B$ domain $Ca^{2+}$-binding loops have milder effects in reconstitution assays and do not affect vesicle priming, but enhance or impair $Ca^{2+}$-evoked release, respectively. The phenotypes caused by combining these mutations are dominated by the K603E and R769E mutations. Our results show that the $C_1$-$C_2B$ region of Munc13-1 plays a central role in vesicle priming and support the notion that two distinct faces of this region control neurotransmitter release and short-term presynaptic plasticity.

## Editor's evaluation

Munc13 is essential for exocytotic fusion of synaptic vesicles but the precise mechanism of action of this multidomain protein are not fully understood. The authors show that two different structural states of Munc13 are involved in, $Ca^{2+}$-independent and $Ca^{2+}$-dependent, synaptic vesicle priming. This is a significant contribution furthering our understanding of the complex multiprotein machinery responsible for the final steps in vesicle exocytosis. The study is comprehensive, careful, using a battery of different approaches that all substantiate the main conclusions of the work.

## Introduction

The release of neurotransmitters by $Ca^{2+}$-evoked synaptic vesicle exocytosis is a central event in neuronal communication. This process involves a series of steps that include tethering of synaptic vesicles (SVs) to specialized areas of the plasma membrane called active zones, priming of the vesicles to a release-ready state and $Ca^{2+}$-triggered fusion of the vesicle and plasma membranes (*Südhof, 2013*). Neurotransmitter release does not merely constitute a means to transmit signals between neurons. The efficiency of release is regulated by a wide variety of mechanisms during presynaptic plasticity processes that shape the properties of neural networks and underlie multiple forms of information processing in the brain (*Regehr, 2012*). Hence, presynaptic terminals can be viewed as minimal computational units of the brain, and understanding the mechanisms that modulate neurotransmitter release is crucial to understand brain function.

The sophisticated protein machinery that controls neurotransmitter release has been extensively characterized (*Brunger et al., 2018*; *Rizo, 2018*), yielding defined models for the functions of the central components of this machinery and allowing reconstitution of fundamental features of synaptic exocytosis in liposome fusion assays that reproduce the critical functional importance of each one of these components (*Ma et al., 2013*; *Stepien and Rizo, 2021*). The SNAP receptor (SNARE) proteins syntaxin-1, SNAP-25 and synaptobrevin play a key role in membrane fusion by forming a tight SNARE complex that consists of a four-helix bundle and brings the vesicle and plasma membranes together (*Hanson et al., 1997*; *Poirier et al., 1998*; *Söllner et al., 1993*; *Sutton et al., 1998*). N-ethyl maleimide sensitive factor (NSF) and soluble NSF attachment proteins (SNAPs) disassemble the cis-SNARE complexes that result after fusion, recycling the SNAREs (*Mayer et al., 1996*; *Söllner et al., 1993*). NSF and SNAPs also dissociate trans-SNARE complexes and other four-helix bundles formed by the SNAREs (*Choi et al., 2018*; *Ma et al., 2013*; *Prinslow et al., 2019*; *Yavuz et al., 2018*), thus hindering fusion mediated by the SNAREs alone but at the same time ensuring that release occurs through a highly regulated fusion pathway that requires Munc18-1 and Munc13s (*Park et al., 2017*; *Stepien et al., 2019*), and that ensures proper parallel assembly of the SNARE four-helix bundle (*Lai et al., 2017*). Munc18-1 and Munc13s are essential for neurotransmitter release (*Augustin et al., 1999*; *Richmond et al., 1999*; *Varoqueaux et al., 2002*; *Verhage et al., 2000*) and organize trans-SNARE complex assembly via an NSF-SNAP-resistant pathway that starts with Munc18-1 bound to a self-inhibited 'closed' conformation of syntaxin-1 (*Dulubova et al., 1999*; *Misura et al., 2000*). Munc18-1 also binds to synaptobrevin, forming a template to assemble the SNARE complex (*Baker et al., 2015*; *Jiao et al., 2018*; *Parisotto et al., 2014*; *Sitarska et al., 2017*) while Munc13-1 bridges the vesicle and plasma membranes (*Liu et al., 2016*; *Quade et al., 2019*) and helps to open syntaxin-1 (*Ma et al., 2011*; *Yang et al., 2015*). Synaptic vesicle fusion upon $Ca^{2+}$ influx is triggered by the $Ca^{2+}$ sensor synaptotagmin-1 (*Fernández-Chacón et al., 2001*).

Munc13s act as master regulators of release through their multidomain architecture. Munc13-1, the major brain isoform, includes a variable N-terminal region and a C-terminal region that is conserved in all Munc13 isoforms (*Figure 1A*). The N-terminal region contains a calmodulin-binding region involved in $Ca^{2+}$-dependent short-term plasticity (*Junge et al., 2004*) and a $C_2A$ domain that forms a homodimer as well as a heterodimer with αRIMs, coupling RIM-dependent forms of plasticity to the priming machinery (*Betz et al., 2001*; *Camacho et al., 2017*; *Deng et al., 2011*; *Dulubova et al., 2005*; *Lu et al., 2006*). The conserved C-terminal region of Munc13-1 includes: (i) the $C_1$ domain, which mediates diacylglycerol (DAG)/phorbol ester-dependent potentiation of release (*Basu et al., 2007*; *Rhee et al., 2002*); (ii) the $C_2B$ domain, which binds $Ca^{2+}$ and phosphatidylinositol 4,5-bisphosphate ($PIP_2$) and is involved in $Ca^{2+}$-dependent short-term plasticity (*Shin et al., 2010*); (iii) the MUN domain, which facilitates syntaxin-1 opening (*Ma et al., 2011*; *Magdziarek et al., 2020*); and (iv) the $C_2C$ domain, which binds weakly to membranes in a $Ca^{2+}$-independent manner (*Quade et al., 2019*). The crystal structure of a Munc13-1 fragment containing the $C_1$, $C_2B$ and MUN domains (*Xu et al., 2017*) revealed that the $C_1$ and $C_2B$ domains pack at one end of the highly elongated MUN domain, with their DAG- and $Ca^{2+}/PIP_2$-binding sites pointing to the same direction (*Figure 1A*). The $C_2C$ domain [illustrated by a homology model of the $C_2C$ domain (*Quade et al., 2019*) in *Figure 1A*] is expected to emerge on the opposite end of the MUN domain. This architecture enables the membrane bridging activity of Munc13-1, which led to a model whereby the $C_2C$ domain binds to the vesicle membrane while the $C_1$-$C_2B$ region binds to the plasma membrane (*Liu et al., 2016*; *Quade et al., 2019*; *Xu et al., 2017*). This model was supported by the finding

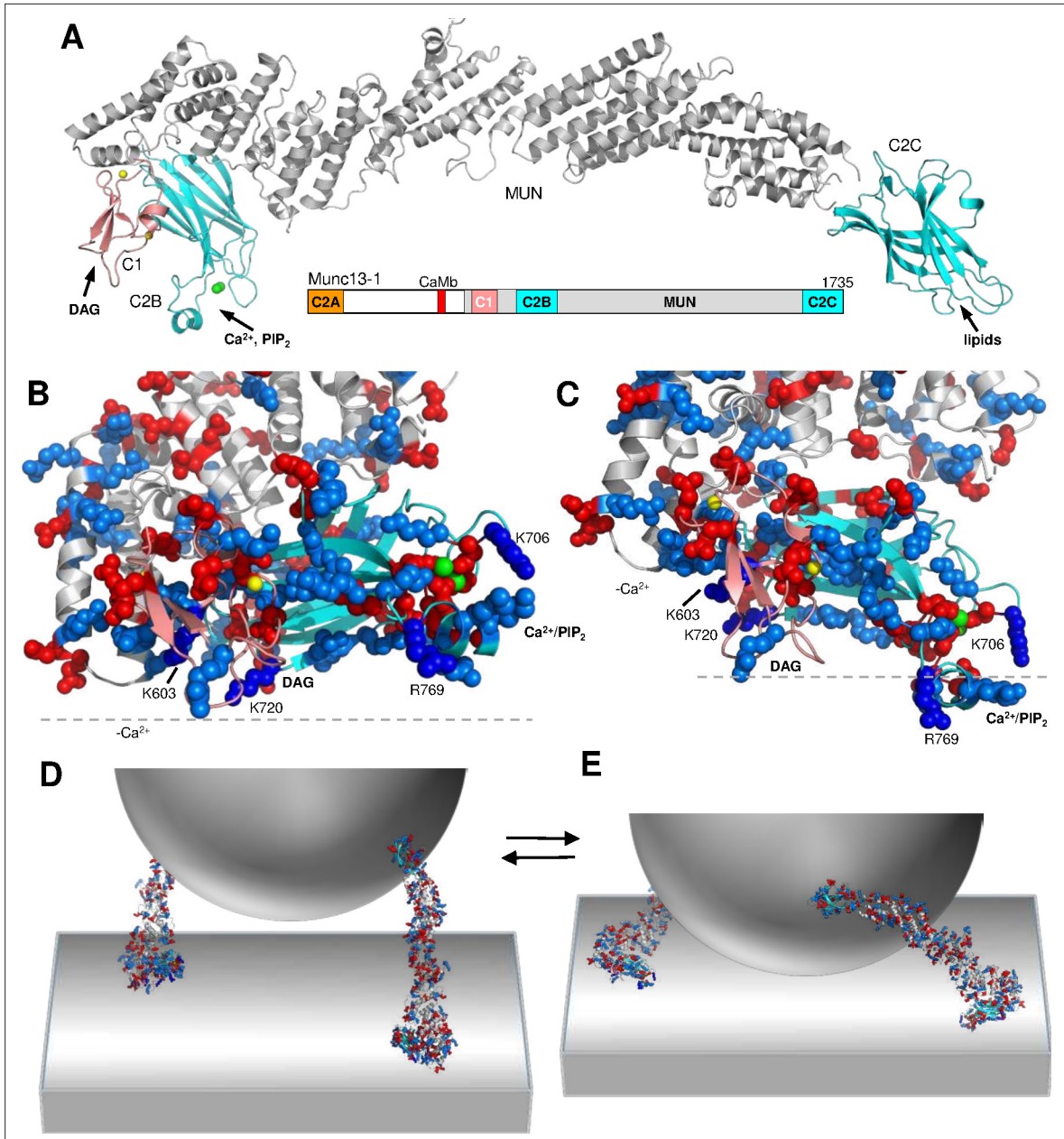

**Figure 1.** Model of Munc13-1 function with two different orientations bridging the membranes. (**A**) Domain diagram of Munc13-1 (CaMb, calmodulin binding domain) and ribbon diagrams illustrating a model of the structure of the Munc13-1 $C_1C_2BMUNC_2C$ fragment. The ribbon diagram is based on the crystal structure of the Munc13-1 $C_1C_2BMUN$ fragment (***Xu et al., 2017***) (PDB accession number 5UE8) but, because some residues in loops of the $C_1$ and $C_2B$ domains of this structure are missing, these domains were replaced with the NMR structure of the $C_1$ domain (***Shen et al., 2005***) (salmon, PDB accession number 1Y8F) and the crystal structure of the $Ca^{2+}$-bound $C_2B$ domain (***Shin et al., 2010***) (cyan, PDB accession number 3KWU). The cyan ribbon diagram on the right is a model of the $C_2C$ domain built by homology with the RIM1$\alpha$ $C_2B$ domain (***Guan et al., 2007***) (PDB accession code 2Q3X). (**B,C**) Closeup views of the structure of the model of the Munc13-1 $C_1C_2BMUN$ fragment shown in (**A**). The dashed lines indicate the positions of planes parallel to the plasma membrane in the orientations predicted when binding is mediated by the polybasic face (**B**) or the DAG/$Ca^{2+}$/$PIP_2$-binding face (**C**) of the $C_1$-$C_2B$ region. The $Zn^{2+}$ ions bound to the $C_1$ domain and the $Ca^{2+}$ ions bound to the $C_2B$ domain are shown as yellow or green spheres, respectively. Basic residues are shown as blue spheres and acidic residues as red spheres. The positions of the residues that were mutated in this study and the approximate locations of the DAG- and $Ca^{2+}$/$PIP_2$-binding sites are indicated. (**D,E**) Three-dimensional models that illustrate the notion that Munc13-1 $C_1C_2BMUNC_2C$ can bridge the synaptic vesicle and plasma membranes in two different orientations. The models include a vesicle (half sphere), the plasma membrane (flat surface) and ribbon diagrams representing the modelled structure of the $C_1C_2BMUNC_2C$ fragment shown in (**A**) in the approximately perpendicular (**D**) and slanted (**E**) orientations with respect to the plasma membrane yielded by the MD simulations performed in the absence and presence of $Ca^{2+}$, respectively. The model postulates that perpendicular orientation allows partial but not full assembly of the SNARE

*Figure 1 continued on next page*

*Figure 1 continued*

complex, whereas the slanted orientation facilitates full assembly. The two states are proposed to exist in an equilibrium that is shifted to the right by $Ca^{2+}$ and DAG.

The online version of this article includes the following figure supplement(s) for figure 1:

**Figure supplement 1.** Representative conformations showing the orientations of the $C_1C_2BMUNC_2C$ fragment with respect to the flat bilayer visited during the MD simulations.

**Figure supplement 2.** Closeup views of the membrane-binding modes of the $C_1C_2BMUN$ fragment observed in the final frames of the MD simulations performed in the absence (**A**) or presence (**B**) of $Ca^{2+}$.

that mutations in predicted membrane-binding residues of the $C_2C$ domain dramatically impaired the liposome clustering ability of a fragment spanning the Munc13-1 $C_1$, $C_2B$, MUN and $C_2C$ domains ($C_1C_2BMUNC_2C$) in vitro, as well as vesicle docking and neurotransmitter release in neurons (*Quade et al., 2019*).

The advances made in understanding the core of the release machinery provide a framework to elucidate the molecular mechanisms that underlie Munc13-1-dependent presynaptic plasticity. A model of how increases in DAG and intracellular $Ca^{2+}$ levels during repetitive stimulation facilitate vesicle priming and neurotransmitter release emerged from the crystal structure of a Munc13-1 $C_1C_2BMUN$ fragment and the realization that the $C_1$ and $C_2B$ domains could bind to the plasma membrane through their DAG- and $Ca^{2+}/PIP_2$-binding sites in a $Ca^{2+}$-dependent manner, and through a polybasic region that partially overlaps with these sites in the absence of $Ca^{2+}$ (referred to below as the polybasic face; see *Figure 1B and C*; *Xu et al., 2017*). These observations, together with liposome clustering and liposome fusion data, suggested that Munc13-1 can bridge a vesicle to the plasma membrane in two orientations (*Liu et al., 2016*; *Quade et al., 2019*; *Xu et al., 2017*): a close to perpendicular orientation that involves the polybasic face of the $C_1$-$C_2B$ region is favored in the absence of $Ca^{2+}$ and facilitates partial assembly of SNARE complexes but hinders progress toward membrane fusion; and a more slanted orientation that also involves the $C_1$-$C_2B$ region is favored by $Ca^{2+}$ and DAG, and stimulates further SNARE complex assembly and fusion (*Figure 1D and E*). Interestingly, a large amount of experimental data on short-term presynaptic plasticity accumulated over the years can be explained by a related model whereby there is a dynamic equilibrium between two primed states that involve different orientations of Munc13-1 and different extents of SNARE complex assembly, and the equilibrium can be shifted by $Ca^{2+}$ and DAG (*Neher and Brose, 2018*). The proposed dual inhibitory and stimulatory roles of the membrane bridging activity of Munc13-1 are also consistent with the effects of point mutations that disrupt interactions between the various Munc13-1 domains (*Xu et al., 2017*), with the finding that a H567K mutation that unfolds the $C_1$ domain of Munc13-1 impairs vesicle priming but increases the vesicular release probability (*Basu et al., 2007*; *Rhee et al., 2002*), and with the observation that deletion of the $C_1$ or $C_2B$ domain of unc-13 enhances neurotransmitter release in *C. elegans*, yet deletion of both domains strongly hinders release (*Michelassi et al., 2017*). Nevertheless, the validity of this two-state model and the functional importance of the polybasic face formed by the $C_1$ and $C_2B$ domains remain to be demonstrated.

The study presented here was designed to investigate the functional consequences of mutations in predicted membrane-binding residues of the Munc13-1 $C_1$-$C_2B$ region and test this model using a combination of molecular dynamics (MD) simulations, biochemical and reconstitution assays in vitro, and electrophysiological experiments in neurons. We find that K603E or R769E single point mutations in the $C_1C_2B$ polybasic face of Munc13-1 disrupt $Ca^{2+}$-independent liposome binding and bridging, as well as stimulation of liposome fusion in reconstitution assays, and severely impair synaptic vesicle priming in mice autaptic cultures. Conversely, a K720E mutation in the polybasic face and a K706E mutation in the $C_2B$ domain $Ca^{2+}$-binding loops, which have milder effects on liposome binding, bridging and fusion, do not affect vesicle priming; however, $Ca^{2+}$-evoked release and the release probability are enhanced by the K720E mutation and impaired by K706E. The phenotypes caused by combining these mutations are dominated by the K603E and R769E mutations. These results strongly support the notion that binding of the $C_1$-$C_2B$ region of Munc13-1 to the plasma membrane is critical for synaptic vesicle priming and that two distinct membrane-binding faces of this region control neurotransmitter release.

## Results

### Models of Munc13-1 $C_1$-$C_2B$-membrane interactions in the absence and presence of $Ca^{2+}$

The notion that the $C_1$-$C_2B$ region of Munc13-1 can bind to membranes via different surfaces in the absence and presence of $Ca^{2+}$ emerged from analysis of the distribution of charged residues in these surfaces (*Xu et al., 2017*; *Figure 1B and C*). To derive models of these putative membrane-binding modes, we performed MD simulations using a model of a fragment spanning the $C_1$, $C_2B$ and MUN domains of Munc13-1 ($C_1C_2BMUN$) based on its crystal structure (*Xu et al., 2017*), and a square membrane with a lipid composition that mimics that of the plasma membrane (*Chan et al., 2012*). In one simulation, we placed the $Ca^{2+}$-free $C_1C_2BMUN$ model above the cytoplasmic leaflet in an approximately perpendicular orientation with the polybasic face close to the membrane, and ran a simulation of 100 ns. The orientation of $C_1C_2BMUN$ became even more perpendicular in the first 10 ns and during the remaining 90 ns oscillated around an almost completely perpendicular orientation such as that observed at the end of the simulation (*Figure 1—figure supplement 1A,B*). The $C_1$-$C_2B$ region quickly established extensive interactions with the membrane, including multiple salt bridges between basic residues and the phospholipid head groups (*Figure 1—figure supplement 2A*).

We carried a second simulation in which we included two $Ca^{2+}$ ions bound to the corresponding binding sites of the $C_2B$ domain (*Shin et al., 2010*) and $C_1C_2BMUN$ was placed in a more slanted orientation such that the DAG- and $Ca^{2+}$/$PIP_2$-binding sites of the $C_1$ and $C_2B$ domains, respectively, were close to the membrane. A range of slanted orientations of $C_1C_2BMUN$ were visited during the 86 ns simulation, oscillating during the last 70 ns around the orientation observed in the last pose, which was even more slanted than the initial orientation (*Figure 1—figure supplement 1C,D*). During the simulation, the expected DAG and $PIP_2$ binding regions of the $C_1$ and $C_2B$ domains became intimately bound to the membrane through numerous interactions involving hydrophobic and basic residues of both domains (*Figure 1—figure supplement 2B*). Interestingly, the unique amphipathic α-helix present in one of the $Ca^{2+}$-binding loops of the $C_2B$ domain (*Shin et al., 2010*) inserted partially into the membrane through its hydrophobic side chains while its basic residues interacted with the phospholipid head groups. It is noteworthy that this helix also interacts with the membrane in the perpendicular orientation adopted in the $Ca^{2+}$-free simulation, which allows a more extensive membrane-interacting surface involving the sides of the $C_1$ and $C_2B$ domains, as well as the linker between them (*Figure 1—figure supplements 1B and 2A*). In the slanted, $Ca^{2+}$-bound orientation, the interaction surface is smaller because it involves only the tips of the $C_1$ and $C_2B$ domain, but hydrophobic groups of the $C_2B$ domain helix and the $C_1$ domain insert into the membrane, and a $PIP_2$ head group located between the $Ca^{2+}$-binding loops is close to the $Ca^{2+}$ ions (*Figure 1—figure supplements 1D and 2B*). Since phospholipid head groups are known to complete the coordination spheres of $Ca^{2+}$ ions bound to $C_2$ domains, dramatically increasing their $Ca^{2+}$ affinity (*Verdaguer et al., 1999*; *Zhang et al., 1998*), it seems likely that phospholipid-$Ca^{2+}$ coordination and the insertion of hydrophobic groups into the membrane provide the driving force for this binding mode and a $Ca^{2+}$-induced change in orientation of Munc13-1 with respect to the membrane.

We would like to emphasize that these simulations were biased by the initial orientations and a much more thorough analysis will be required to explore the possible binding modes of the Munc13-1 $C_1$-$C_2B$ region with the plasma membrane. However, these short simulations were dynamically stable around the two distinct basins, with and without $Ca^{2+}$, supporting the notion that Munc13-1 can adopt two dramatically different orientations with respect to the plasma membrane. The simulations yielded chemically reasonable binding modes and help to visualize how changes in the orientation of Munc13-1 can drastically alter the distance between a vesicle and the plasma membrane (*Figure 1D and E*). Note also that, although we postulate that $Ca^{2+}$ favors the switch from the perpendicular to the slanted orientation, both binding modes are likely possible in the absence and presence of $Ca^{2+}$, and other factors such as a more complete assembly of the SNARE complex may induce the switch to the slanted orientation even in the absence of $Ca^{2+}$. The observed binding modes provide useful frameworks for the design of mutations to disrupt Munc13-1-membrane interactions involving the $C_1C_2B$ region. Using these models, we designed the following mutations to investigate the functional importance of the distinct faces of the $C_1$-$C_2B$ region and test the hypothesis that this region mediates binding of Munc13-1 to membranes in two different orientations that underlie in part its role in regulating neurotransmitter release: (i) K603E and K720E mutations in the $C_1$ and $C_2B$ domain, respectively,

probed residues in the polybasic face that are far from the $Ca^{2+}$-binding sites of the $C_2B$ domain and are predicted to interact with the lipids in the $Ca^{2+}$-independent binding mode (*Figure 1B*, *Figure 1—figure supplement 2A*); (ii) an R769E mutation probed the importance of a basic residue from one of the $Ca^{2+}$-binding loops of the $C_2B$ domain that is involved in interactions with the lipids in both the $Ca^{2+}$-independent and $Ca^{2+}$-dependent binding modes yielded by the simulations (*Figure 1B and C*, *Figure 1—figure supplement 2A, B*); and (iii) a K706E mutation in a residue from another $Ca^{2+}$-binding loop of the $C_2B$ domain (loop 1) that is involved in lipid interactions in the $Ca^{2+}$-dependent membrane binding mode but is far from the membrane in the $Ca^{2+}$-independent mode (*Figure 1B and C*, *Figure 1—figure supplement 2A, B*). Note that a mutation of this residue to Trp in ubMunc13-2 led to a gain-of-function, increasing the $Ca^{2+}$ sensitivity of neurotransmitter release (*Shin et al., 2010*).

## Mutations in the Munc13-1 $C_1$-$C_2B$ polybasic region impair membrane binding

The proposal that $C_1C_2BMUNC_2C$ can bridge membranes in two different orientations (*Figure 1D and E*) emerged not only from the structural studies of the $C_1C_2BMUN$ fragment that revealed the polybasic face of the $C_1$-$C_2B$ region (*Xu et al., 2017*), but also from liposome clustering assays and reconstitution experiments with liposomes containing synaptobrevin (V-liposomes) and liposomes containing syntaxin-1 and SNAP-25 (T-liposomes). Thus, fusion of these liposomes in the presence of Munc18-1, NSF and αSNAP requires $Ca^{2+}$ and the Munc13-1 $C_1C_2BMUNC_2C$ fragment, and depends on the ability of $C_1C_2BMUNC_2C$ to bridge the liposome membranes, but the liposome clustering activity of $C_1C_2BMUNC_2C$ is comparable in the absence and presence of $Ca^{2+}$, indicating that $Ca^{2+}$ induces a switch to an active orientation required for fusion (*Liu et al., 2016*; *Quade et al., 2019*). Mutagenesis studies demonstrated the key importance of the $C_2C$ domain for the ability of $C_1C_2BMUNC_2C$ to bridge membranes and support liposome fusion (*Quade et al., 2019*), but the role of the $C_1C_2B$ region has not been tested. To address this question and test the relevance of the binding modes observed in our MD simulations, we used a combination of assays that measure liposome binding, clustering or fusion, and analyzed the effects of single K603E, K706E, K720E, and R769E mutations in these assays. In addition, we studied the effects caused by double mutations that combined the two point mutations in residues that are far from the $C_2B$ domain $Ca^{2+}$-binding sites (K603E/K720E) or the two mutations in the $C_2B$ domain $Ca^{2+}$-binding loops (K706E/R769E), as well as a quadruple mutant where the four basic residues were replaced (K603E/K720E/K706E/R769E).

We first tested the effects of these mutations on binding to protein-free liposomes with the lipid composition that we normally used for T-liposomes, which mimics that of the plasma membrane (*Ma et al., 2013*), using liposome co-sedimentation assays. All mutations decreased $Ca^{2+}$-independent liposome binding to some degree compared to wild type (WT) $C_1C_2BMUNC_2C$ (top panels of *Figure 2A and B*, *Figure 2—source data 1*), which may arise in part because they all decrease the overall positive electrostatic potential in the region. Among the single point mutations, K603E and R769E impaired binding most strongly, while K720E had an intermediate effect and K706E exhibited the weakest effect on binding. $Ca^{2+}$-independent liposome binding was strongly disrupted by the two double mutations and the quadruple mutation. These results are consistent with the $Ca^{2+}$-independent binding mode predicted for the $C_1$-$C_2B$ region (*Figure 1B*, *Figure 1—figure supplement 2A*), with a natural variation in the energetic contributions of the residues of the polybasic face to binding that is manifested in the different impairment caused by the K720E mutation compared with K603E and R769E.

$Ca^{2+}$-dependent liposome binding was not substantially decreased by any of the single mutations or by the K603E/K720E double mutation. The K706E/R769E mutation appeared to impair $Ca^{2+}$-dependent liposome binding to a moderate degree, while binding was almost abrogated by the quadruple mutation (*Figure 2A*, middle panel, and 2B, lower panel). These results suggest that $Ca^{2+}$ substantially increases the affinity of $C_1C_2BMUNC_2C$ for the liposomes. Such an increase may not be observable for WT $C_1C_2BMUNC_2C$ because under the conditions of our experiments binding is likely strong enough for saturation in the absence and presence of $Ca^{2+}$, but $Ca^{2+}$-induced enhancement of binding becomes detectable when the mutations decrease the overall affinity. For the same reason, the effects of single mutations on $Ca^{2+}$-dependent liposome binding may not be detectable even if the mutated residues contribute to binding, but the effects observed for the double mutants support the notion that K706 and R769 contribute more to $Ca^{2+}$-dependent binding than K603 and K720,

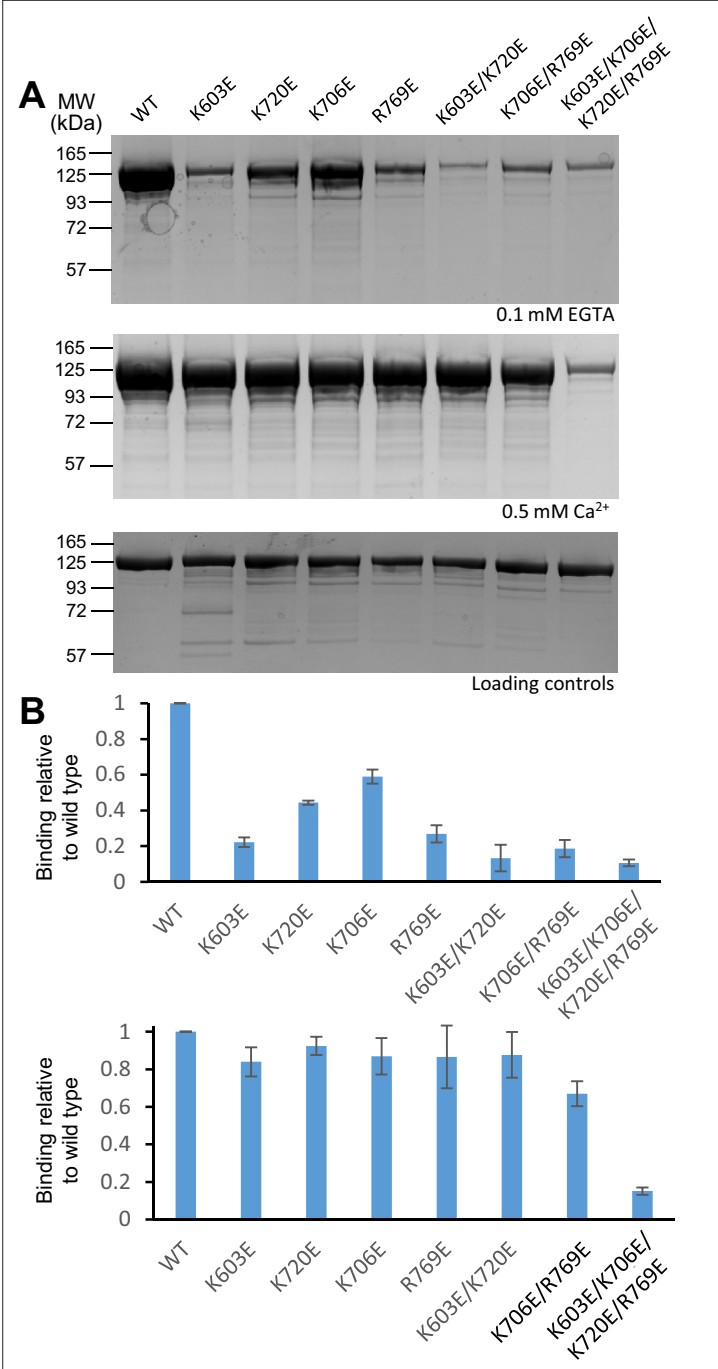

**Figure 2.** Mutations in basic residues of the $C_1$-$C_2B$ region differentially impair the liposome affinity of the $C_1C_2BMUNC_2C$ fragment. (**A**) Liposome co-sedimentation assays were performed with WT and mutant Munc13-1 $C_1C_2BMUNC_2C$ fragments and the pellets were analyzed by SDS-PAGE followed by coomassie blue staining. The top and middle images show experiments performed in the presence of 0.1 mM EGTA or 0.5 mM $Ca^{2+}$, respectively. The bottom panel shows loading controls. The positions of molecular weight markers are indicated on the left. The total amount of $C_1C_2BMUNC_2C$ used for each sample of the co-sedimentation assays was 10 µg. Loading controls contained 2 µg of protein. (**B**) Quantification of the relative amount of liposome binding of the mutant $C_1C_2BMUNC_2C$ fragments with respect to WT $C_1C_2BMUNC_2C$. The bands of WT and mutant $C_1C_2BMUNC_2C$ fragments in gels from three independent experiments performed in the presence of 0.1 mM EGTA or 0.5 mM $Ca^{2+}$ were quantified with ImageJ and normalize to the average value obtained for WT $C_1C_2BMUNC_2C$. Bars indicate average values and bars show standard deviations.

*Figure 2 continued on next page*

*Figure 2 continued*
The online version of this article includes the following figure supplement(s) for figure 2:
**Source data 1.** Uncropped gel of *Figure 2A* and quantification of liposome binding (*Figure 2B*).

consistent with the model of *Figure 1C*, *Figure 1—figure supplement 2B*. The strong effect caused by the quadruple mutation indicates that the latter residues also contribute to $Ca^{2+}$-dependent liposome binding, likely by decreasing the overall electrostatic potential in the region.

We next analyzed the ability of the various mutants to cluster protein-free liposomes with the same lipid compositions of V- and T-liposomes using dynamic light scattering (DLS). *Figure 3* compares the particle size distributions observed for a mixture of the V- and T-liposomes alone (black bars) with those observed for the same mixtures in the presence of WT or mutant $C_1C_2BMUNC_2C$ and the absence (blue bars) or presence (red bars) of $Ca^{2+}$. The population weighted average radius calculated under each condition (*Figure 3—figure supplement 1*) provides an idea of the overall amount of liposome clustering, but note that these averaged radii need to be interpreted with caution because of the difficulty in accurately calculating populations of very large particles by this method. As observed previously (*Liu et al., 2016*), WT $C_1C_2BMUNC_2C$ caused strong liposome clustering, and the amount of clustering was comparable in the absence and presence of $Ca^{2+}$ (*Figure 3A*, *Figure 3—source data 1*). In the absence of $Ca^{2+}$, the liposome clustering activity was severely impaired by the single K603E and R769E mutations, the two double mutations and the quadruple mutation, whereas the single K706E and K720E mutants still supported robust liposome clustering (*Figure 3B–H*). In contrast, all the mutants except the quadruple mutant induced substantial liposome clustering in the presence of $Ca^{2+}$.

These results reveal that the liposome clustering activity is affected more strongly by the mutations in the absence than in the presence of $Ca^{2+}$, and generally correlate with the liposome binding data. Some differences in strengths of the effects caused by the mutations in the two assays can be attributed to the fact that clustering involves cooperativity between $C_1C_2BMUNC_2C$ molecules that bridge the same liposome pair or liposomes in the same cluster. Overall, these data show that the $C_1$-$C_2B$ region is critical for the liposome bridging activity of Munc13-1 $C_1C_2BMUNC_2C$ and support the notion that distinct membrane binding modes of the $C_1$-$C_2B$ region mediate membrane bridging in the absence and presence of $Ca^{2+}$, as predicted from the models of *Figure 1B–E*.

## Mutations in the Munc13-1 $C_1$-$C_2B$ polybasic region impair liposome fusion

We next turned to our reconstitution assays in which fusion between V- and T-liposomes in the presence of Munc18-1, Munc13-1 $C_1C_2BMUNC_2C$, NSF and αSNAP is assessed by simultaneously monitoring lipid and content mixing (*Liu et al., 2016*). The strict $Ca^{2+}$ dependence of fusion in these assays arises because of $Ca^{2+}$ binding to the Munc13-1 $C_2B$ domain. The two-state model postulates that such binding favors a more slanted orientation of $C_1C_2BMUNC_2C$, accelerating trans-SNARE complex formation and fusion, and that this phenomenon underlies at least in part the facilitation of neurotransmitter release due to accumulation of intracellular $Ca^{2+}$ during repetitive stimulation (*Shin et al., 2010*). To test whether the $Ca^{2+}$ concentration required to activate the Munc13-1 $C_2B$ domain in the liposome fusion assays is consistent with this proposal, we monitored lipid mixing between V- and T-liposomes in the presence of WT Munc18-1, Munc13-1 $C_1C_2BMUNC_2C$, NSF and αSNAP as a function of $Ca^{2+}$ concentration. The fluorescence of the $Ca^{2+}$-sensing dye Fluo-4 was used to measure the $Ca^{2+}$ concentration in each point of the titration, and lipid mixing between the liposomes was monitored simultaneously from the de-quenching of DiD-lipids present in the V-liposomes. We observed a steep dependence of lipid mixing on the $Ca^{2+}$ concentration (*Figure 4A*, *Figure 4—source data 1*), and fitting plots of the lipid mixing observed at 1,500 or 700 s as a function of $Ca^{2+}$ concentration to a Hill equation yielded EC50s of 584 nM and 966 nM, respectively (*Figure 4B*). These data show that the Munc13-1 $C_2B$ domain is activated in these fusion assays at submicromolar $Ca^{2+}$ concentrations, comparable to those expected to accumulate near $Ca^{2+}$ channels in a presynaptic active zone during repetitive stimulation.

We then examined the effects of the mutations in basic residues of the $C_1$-$C_2B$ region of Munc13-1 on liposome fusion using these assays, in which the V- and T-liposomes commonly have

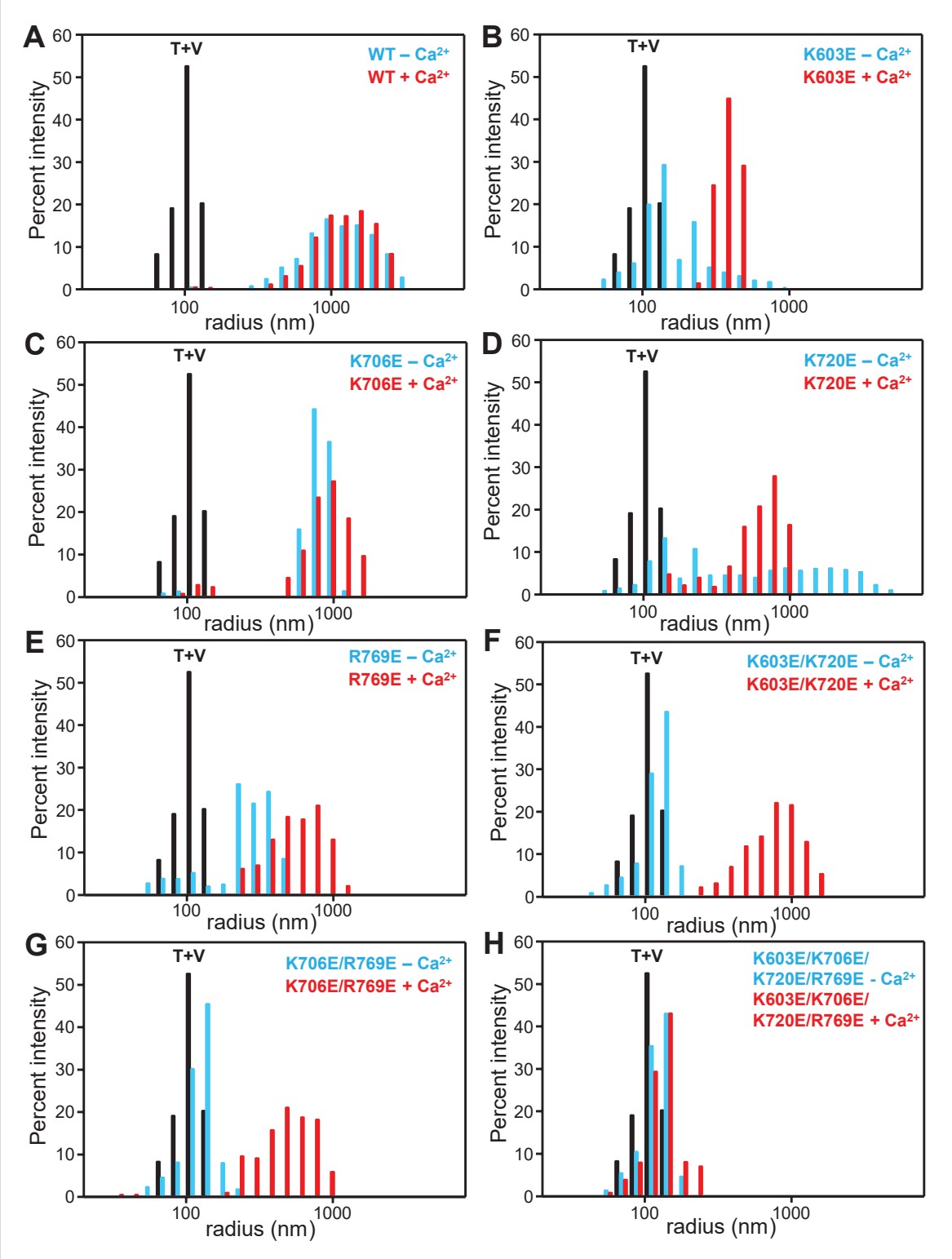

**Figure 3.** Mutations in basic residues of the $C_1$-$C_2B$ region differentially impair the ability of the $C_1C_2BMUNC_2C$ fragment to cluster V- and T-liposomes. The ability of WT and mutant Munc13-1 $C_1C_2BMUNC_2C$ fragments to cluster V- and T-liposomes was analyzed by DLS. The diagrams show the particle size distributions observed in mixtures of the V- and T-liposomes alone (T + V, black bars), after addition of the corresponding $C_1C_2BMUNC_2C$ fragment in the presence of 0.1 mM EGTA (blue bars), and after addition of 0.6 mM $Ca^{2+}$ to the same sample (red bars).

The online version of this article includes the following figure supplement(s) for figure 3:

*Figure 3 continued on next page*

*Figure 3 continued*

**Source data 1.** Percent intensities of radius bins.

**Figure supplement 1.** Semi-quantitative analysis of the liposome clustering assays.

synaptobrevin-to-lipid ratio 1:500 and syntaxin-1-to-lipid ratio 1:800 (*Liu et al., 2016*). $Ca^{2+}$-dependent liposome fusion was not impaired by any of the single point mutations or by the double K603E/K720E mutation (*Figure 5A, B and E*, *Figure 5—source data 1*). However, the double K706E/R769E mutation did impair fusion strongly, and fusion was abrogated by the quadruple K603E/K720E/K706E/R769E mutation (*Figure 5A, B and E*). Thus, K706 and R769, which are in the $Ca^{2+}$-binding loops of the $C_2B$ domain, are critical for $Ca^{2+}$-dependent activation of fusion, while K603 and K720 did not have detectable inhibitory effects. However, it is important to note that two aspects of these assays limit their application to analyze the effects of mutations on fusion. First, inhibitory effects of the mutations on $Ca^{2+}$-independent fusion cannot be analyzed because there is no fusion in the absence of $Ca^{2+}$ in

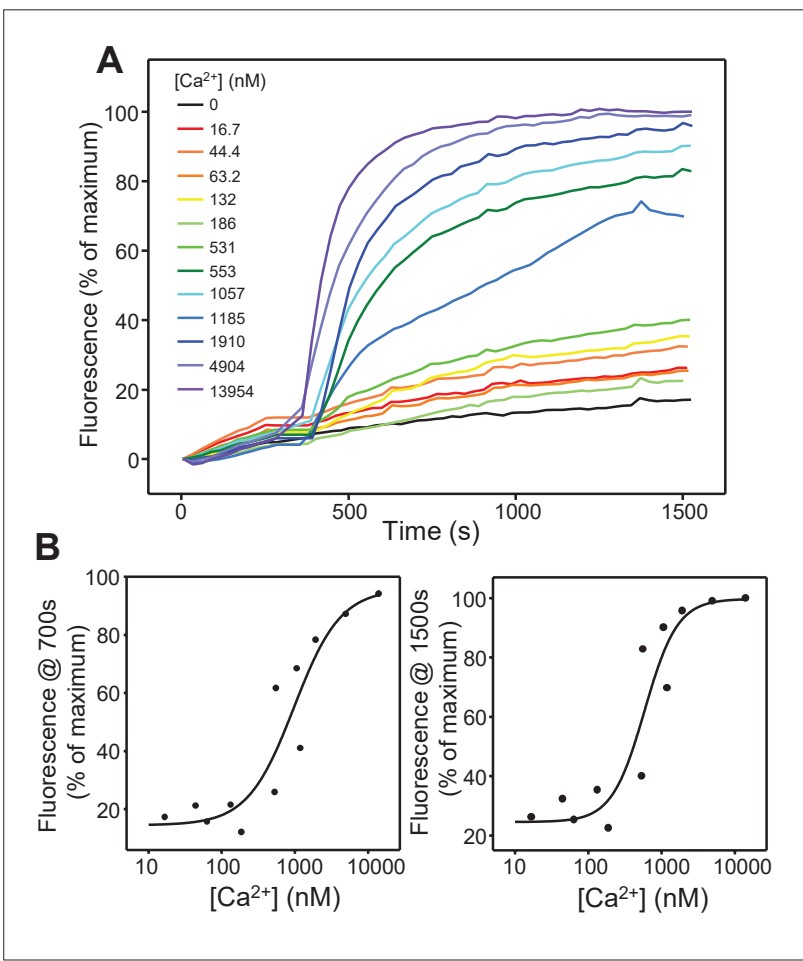

**Figure 4.** $Ca^{2+}$-dependence of the stimulation of liposome fusion by the Munc13-1 $C_1C_2BMUNC_2C$. (**A**) Lipid mixing assays performed at different $Ca^{2+}$ concentrations. Lipid mixing between V- and T-liposomes was measured from the fluorescence de-quenching of DiD lipids in the V-liposomes. The assays were performed in the presence of Munc18-1, NSF, αSNAP, WT Munc13-1 $C_1C_2BMUNC_2C$ fragment and Fluo-4. Experiments were started in the presence of 100 μM EGTA and $Ca^{2+}$ was added at different concentrations after 300 s. The concentration of free $Ca^{2+}$ in each experiment was assessed from the fluorescence of Fluo-4. (**B**) $Ca^{2+}$ dependence of the relative fluorescence at 700 s and 1,500 s of the lipid mixing assays of panel (**A**). The data were fit to a Hill equation.

The online version of this article includes the following figure supplement(s) for figure 4:

**Source data 1.** Time dependence of lipid mixing and $Ca^{2+}$-dependence of the relative fluorescence intensities at 700 s and 1,500 s.

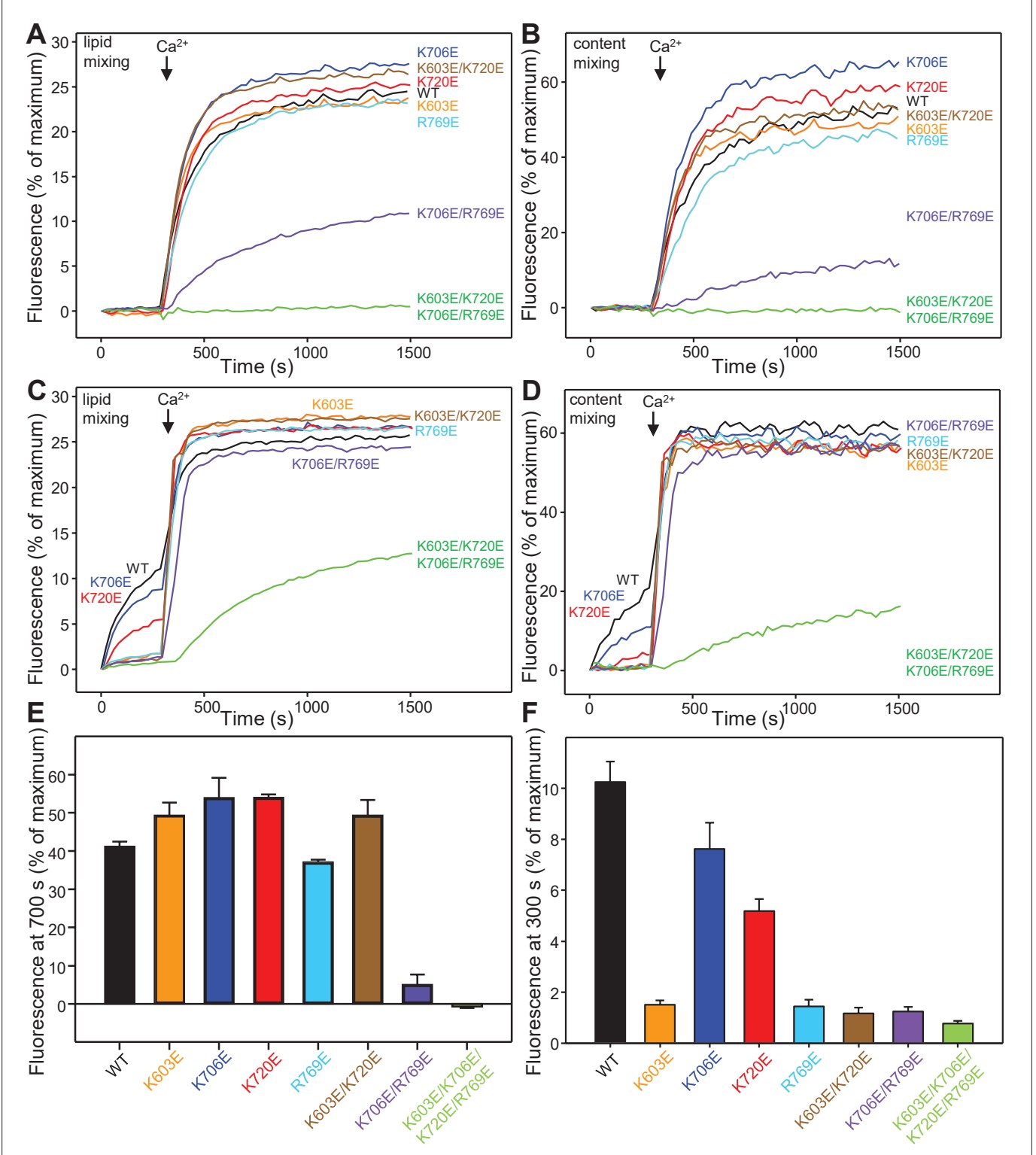

**Figure 5.** Mutations in basic residues of the $C_1$-$C_2B$ region differentially impair the ability of the Munc13-1 $C_1C_2BMUNC_2C$ fragment to stimulate liposome fusion. (**A,B**) Lipid mixing (left) between V- and T-liposomes was measured from the fluorescence de-quenching of Marina Blue-labeled lipids and content mixing (right) was monitored from the development of FRET between PhycoE-Biotin trapped in the T-liposomes and Cy5-Streptavidin trapped in the V-liposomes. The assays were performed in the presence of WT Munc18-1, NSF, αSNAP and the indicated Munc13-1 $C_1C_2BMUNC_2C$ fragments. Experiments were started in the presence of 100 μM EGTA and 5 mM streptavidin, and $Ca^{2+}$ (600 μM) was added after 300 s. (**C,D**) Analogous lipid and content mixing assays performed under the same conditions but using D326K mutant Munc18-1. (**E**) Quantification of the content mixing

*Figure 5 continued on next page*

*Figure 5 continued*

observed after 700 s in reconstitution assays performed under the conditions of Panels (**A,B. F**). Quantification of the lipid mixing observed after 300 s in reconstitution assays performed under the conditions of Panels (**C,D**). In (**E,F**), bars represent averages of the normalized fluorescence observed in experiments performed at least in triplicate, and error bars represent standard deviations.

The online version of this article includes the following source data and figure supplement(s) for figure 5:

**Source data 1.** Lipid and content mixing between V- and T-liposomes in *Figure 5A–D* and relative fluorescence intensities at 700 s in *Figure 5E* or at 300 s in *Figure 5F*.

**Figure supplement 1.** Mutations in basic residues of the $C_1$-$C_2B$ region differentially impair the ability of the Munc13-1 $C_1C_2BMUNC_2C$ fragment to stimulate liposome fusion.

**Figure supplement 1—source data 1.** Lipid and content mixing between V- and T-liposomes in *Figure 5—figure supplement 1A, B* and relative fluorescence intensities at 1,500 s in *Figure 5—figure supplement 1C,D*.

these assays. Second, because fusion with WT $C_1C_2BMUNC_2C$ is so efficient upon addition of $Ca^{2+}$, gain-of-function effects cannot be observed, and moderate inhibitory effects may also be masked.

To analyze the effects of the mutations on $Ca^{2+}$-independent fusion, we performed analogous assays but using Munc18-1 bearing a gain-of-function mutation (D326K) that supports some $Ca^{2+}$-independent liposome fusion (*Sitarska et al., 2017*; *Figure 5C and D*). The $Ca^{2+}$-independent component of fusion, as assessed from the extent of content mixing, was somewhat inhibited by the K706E mutation, was impaired more strongly by the K720E mutation, and was abolished by all other mutations (*Figure 5D*). Since lipid mixing was disrupted to a smaller extent by the mutations and hence yielded a higher dynamic range for quantification, we measured the lipid mixing at 300 s, right before $Ca^{2+}$ addition, in repeat experiments performed under the same conditions (*Figure 5F*). This analysis showed that lipid mixing was strongly impaired by all the mutations except the K706E and K720E mutations, which inhibited lipid mixing moderately.

To address the second issue and assess whether inhibitory or stimulatory effects of the mutations on $Ca^{2+}$-dependent fusion in the experiments of *Figure 5A and B* might not have been observed because of its high efficiency with WT $C_1C_2BMUNC_2C$, we performed additional assays under conditions that we recently developed to render $Ca^{2+}$-dependent fusion less efficient and highly sensitive to the $Ca^{2+}$ sensor synaptotagmin-1 (Syt1) (*Stepien and Rizo, 2021*). In these assays, we monitored fusion between liposomes containing synaptobrevin and Syt1 at 1:10,000 and 1:1,000 protein-to-lipid (P/L) ratios, respectively, with liposomes containing syntaxin-1 and membrane-anchored SNAP-25 at 1:5,000 and 1:800 P/L ratios, respectively. The results that we obtained were similar to those observed under our standard conditions (*Figure 5A, B and E*), as $Ca^{2+}$-dependent fusion was markedly disrupted by the double K706E/R769E mutation, was abrogated by the quadruple K603E/K720E/K706E/R769E mutation, and was not substantially affected by the other mutations (*Figure 5—figure supplement 1*, *Figure 5—figure supplement 1—source data 1*).

Overall, these results strongly support the proposal that binding of the $C_1$-$C_2B$ region of $C_1C_2BMUNC_2C$ to the T-liposomes is crucial for liposome clustering and fusion, and that different binding modes involving this region in the absence and presence of $Ca^{2+}$ underlie the drastic activation of liposome fusion observed in our assays.

## Differential disruption of neurotransmitter release by mutations in the $C_1C_2B$ region of Munc13-1

To investigate the physiological consequences of the K603E, K706E, K720E and R769E mutations in the $C_1C_2B$ region of Munc13-1, we analyzed synaptic responses in autaptic hippocampal cultures from Munc13-1/2 double knockout (DKO) mice rescued with Munc13-1 carrying these point mutations individually. All expressed Munc13-1 mutants exhibited Munc13-1 fluorescence at VGLUT1-positive compartments, showing their presynaptic localization, and had similar expression levels as Munc13-1 WT (*Figure 6—figure supplement 1*, *Figure 6—figure supplement 1—source data 1*). We first analyzed their impact on SV priming as measured by depletion of the readily-release pool (RRP) of vesicles induced by hypertonic solution (*Rosenmund and Stevens, 1996*). We found that DKO neurons rescued with the Munc13-1 bearing the K603E mutation in the $C_1$ domain or the R769E mutation in one of the $C_2B$ domain $Ca^{2+}$-binding loops exhibited a drastically reduced RRP (*Figure 6A*, *Figure 6—source data 1*). However, no significant impact on RRP size was observed for the K720E

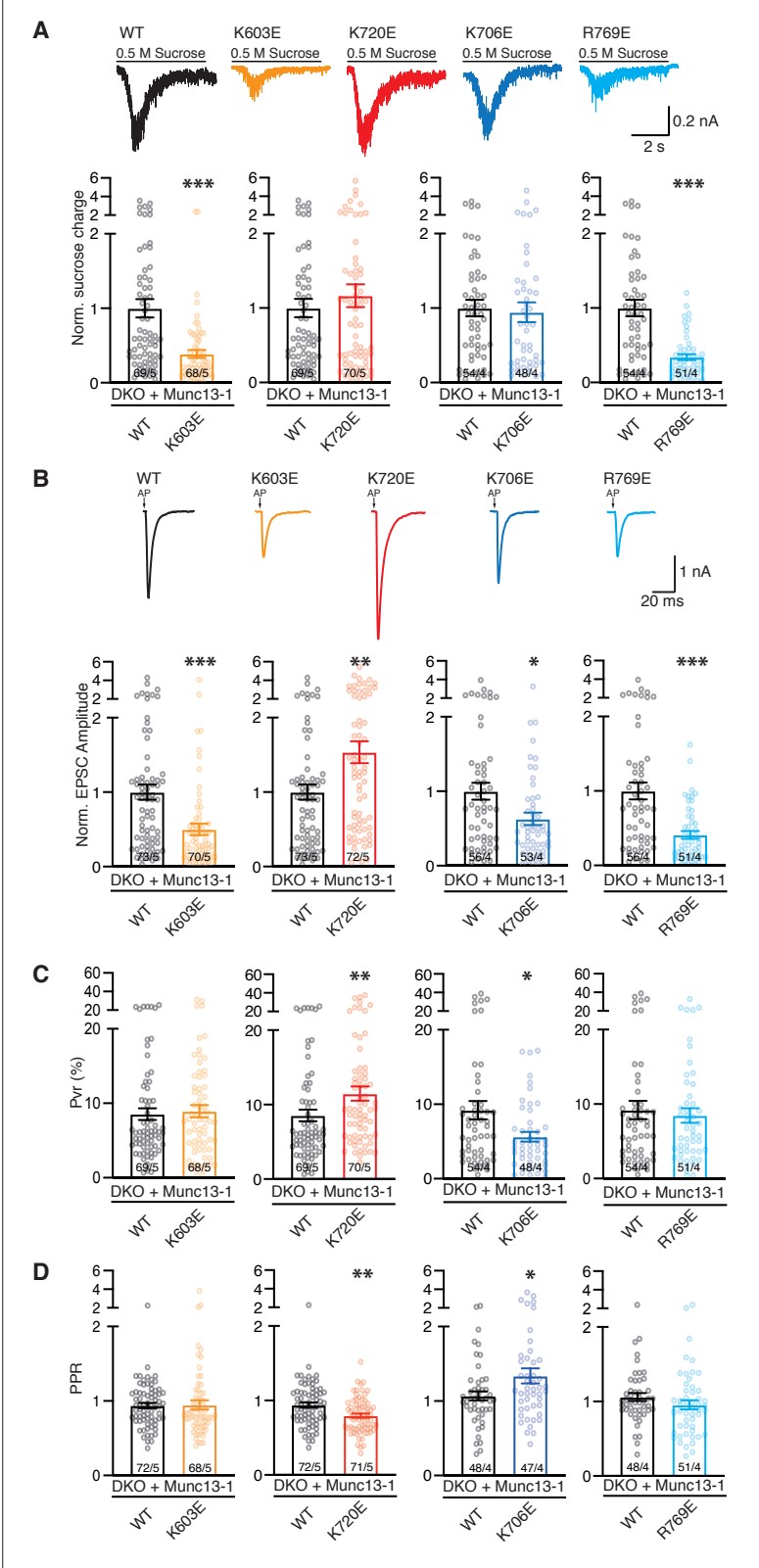

**Figure 6.** Evaluation of Munc13-1 functions using single point mutations within the C1-C2B region. (**A–D**) Electrophysiological parameters of autaptic hippocampal glutamatergic Munc13-1/2 DKO neurons expressing Munc13-1 WT (black) or Munc13-1 polybasic mutants: K603E (orange), K720E (red), K706E (blue), and R769 (light blue). (**A**) Representative sucrose-induced current traces and bar plots depicting the mean charge of the current

*Figure 6 continued on next page*

*Figure 6 continued*

evoked by the application of 500 mM hypertonic sucrose solution for 5 s normalized to the corresponding Munc13-1 WT control. (**B**) Representative traces of AP-evoked EPSCs and bar plots showing the mean EPSC amplitudes normalized to the corresponding Munc13-1 WT. EPSC recordings were done at RT in 2 mM $Ca^{2+}$/4 mM $Mg^{2+}$. Action potentials in traces were blanked for better illustration and substituted by arrows. (**C**) Bar plots depicting the average vesicular release probability (Pvr) calculated for each neuron. (**D**) Bar plots depicting average pair-pulse ratio (PPR) with a 25 ms interpulse interval. Circles in bar plot represent values per neuron. Numbers in bars correspond to the cell number/culture number. Significance and p-values were determined by comparison with the corresponding Munc13-1 WT using the non-parametric Mann-Whitney U test. * $p < 0.05$; ** $p < 0.01$; *** $p < 0.001$.

The online version of this article includes the following figure supplement(s) for figure 6:

**Source data 1.** Normalized sucrose charges, normalized EPSC amplitudes, Pvrs and PPRs in *Figure 6*.

**Figure supplement 1.** Presynaptic expression levels of Munc13-1 $C_1$-$C_2B$ region mutants.

**Figure supplement 1—source data 1.** Fluorescence intensities in *Figure 6—figure supplement 1B*.

**Figure supplement 2.** Quantification of spontaneous neurotransmitter release of point mutations within the Munc13-1 $C_1$-$C_2B$ region.

**Figure supplement 2—source data 1.** Normalized mEPSC frequencies in *Figure 6—figure supplement 2*.

mutation in the $C_2B$ domain, despite the proximity of K720 to K603 (*Figure 1B*), or for the K706E mutation in another $C_2B$ domain $Ca^{2+}$-binding loop (*Figure 6A*). These results clearly correlate with the strong impairment of $Ca^{2+}$-independent liposome binding, clustering and fusion caused by the single K603E and R769E mutations (*Figures 2, 3B, E, 5C, D and F*, *Figure 3—figure supplement 1A*), supporting the notion that the defects in SV priming caused by these mutations arise because they disrupt $Ca^{2+}$-independent interactions of the polybasic face of the $C_1C_2B$ region with the plasma membrane that are critical for Munc13-1 to bridge synaptic vesicles to the plasma membrane.

We next examined the influence of the four mutations on $Ca^{2+}$-evoked neurotransmitter release. The vesicular release probability (Pvr), the likelihood that an action potential causes the fusion of a primed and fusion competent vesicle, can be readily calculated by dividing the excitatory post-synaptic current (EPSC) and sucrose evoked charges (*Reim et al., 2001*; *Rosenmund and Stevens, 1996*). The K603E and R769E mutations decreased $Ca^{2+}$-evoked release but did not alter the Pvr significantly (*Figure 6B and C*). In contrast, the K706E mutation decreased both the EPSC amplitude and the Pvr, while the K720E mutation increased both evoked release and the Pvr (*Figure 6B and C*). These effects on the efficiency of release observed for the K706E and K720E mutants were further confirmed by recordings with a pair pulse protocol consisting of two consecutive AP-induced EPSCs with an inter-stimulus interval of 25 ms (*Figure 6D*). Thus, the K603E and R769E mutations did not alter the paired-pulsed ratio (PPR), but the K706E mutant exhibited an increased PPR, which is typical of synapses with low Pvr, whereas we observed a decreased PPR for the K720E mutant, as expected from its increased Pvr. We also assessed the impact of these mutations on spontaneous release by analyzing the frequency of miniature postsynaptic currents (mEPSCs). We observed a decrease in mEPSC frequency for K603E and R769E, the two mutants that exhibited impaired priming, whereas the K706E mutation did not affect spontaneous release and K720E, the mutant with enhanced Pvr, had increased mEPSC frequency (*Figure 6—figure supplement 2A*, *Figure 6—figure supplement 2—source data 1*). These results show that, while SV priming depends critically on two basic residues in the $C_1C_2B$ region of Munc13-1, K603 and R769, two other basic residues in this region, K706 and K720, modulate the vesicular release probability and have opposite effects.

To gain further insights into the roles of these residues, we analyzed neurotransmitter release in rescue experiments with the Munc13-1 double mutants (K603E/K720E and K706E/R769E), as well as with the quadruple mutant (K603E/K720E/K706E/R769E). The K603E/K720E mutant exhibited impaired sucrose-induced, $Ca^{2+}$-evoked and spontaneous release, without significant changes in Pvr or PPR (*Figure 7*, *Figure 7—source data 1*, *Figure 6—figure supplement 2B*), similar to the effects observed for the single K603E mutant. Hence, the K603E mutation dominates the phenotypes exhibited by this double mutant and cancels the gain-of-function caused by the single K720E mutation. The double point mutant K706E/R769E within the $C_2B$ domain $Ca^{2+}$-binding loops displayed stronger defects in the size of the RRP, the EPSC amplitude and the spontaneous release frequency than the K603E/K720E mutant (*Figure 7A and B*, *Figure 6—figure supplement 2B*), and appeared to have

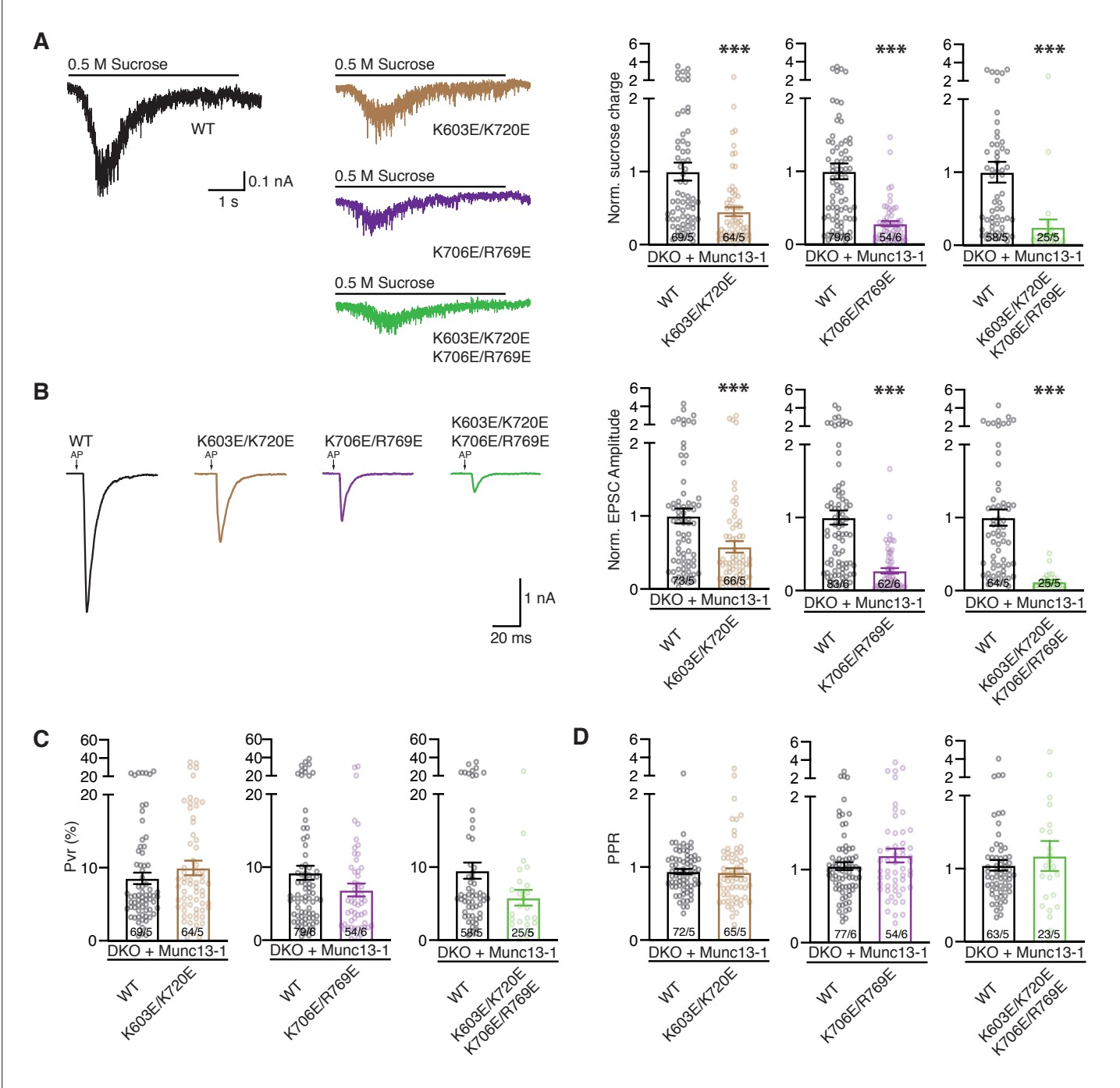

**Figure 7.** Phenotypic analyses of double and quadruple mutations within the Munc13-1 C1-C2B region. (**A**) Representative sucrose-evoked synaptic current traces and bar plots of the charge currents normalized to the corresponding Munc13-1 WT control from autaptic hippocampal glutamatergic Munc13-1/2 DKO neurons rescued with Munc13-1 WT (black), Munc13-1 double point mutants, K603E/K720E (brown) or K706E/R769E (purple), or Munc13-1 quadruple point mutant K603E/K720E/K706E/R769E (green). (**B**) Representative AP-evoked EPSCs traces and bar plots of the normalized mean EPSC amplitudes from the groups above described. Action potentials in traces were blanked and substituted by arrows. (**C**) Bar plots showing the average vesicular release probability (Pvr) from the groups described in A and B. (**D**) Bar plots depicting average PPR at a frequency of 40 Hz. Circles in bar plot represent normalized values per neuron. Numbers in bars corresponded to the cell number/culture number. Significances and p-values were determined using the non-parametric Mann-Whitney U test. *p < 0.05; ***p < 0.001.

The online version of this article includes the following figure supplement(s) for figure 7:

**Source data 1.** Normalized sucrose charges, normalized EPSC amplitudes, Pvrs and PPRs in **Figure 7**.

a tendency to lower Pvr than wild type (WT), but the difference did not reach statistical significance (*Figure 7C*). Correspondingly, the PPR observed for the K706E/R769E mutant showed a tendency to facilitation that was not statistically significant. Finally, we observed that the K603E/K720E/K706E/R769E quadruple mutant exhibited the lowest number of responsive cells and that these were strongly deficient in priming, spontaneous release and, particularly, Ca$^{2+}$-evoked release, while the Pvr and RRP observed for this mutant had similar non-statistically significant tendencies as the K706E/R769E mutant (*Figure 7*, *Figure 6—figure supplement 2B*). These results show that the mutations that impair priming, K603E and R769E, dominate the phenotypes of these double and quadruple mutants and the effects of the mutations that specifically affect evoked release, K706E and K720E, are at least partially masked by these dominant effects.

Overall, these data show that mutations in the C$_1$C$_2$B region of Munc13-1 can have effects on both SV priming and Ca$^{2+}$-evoked release, and can lead to both loss-of-function and gain-of-function. These findings support the notion that the C$_1$C$_2$B region is involved in Ca$^{2+}$-independent interactions through the polybasic face that are critical for priming and in additional interactions involving the C$_2$B domain Ca$^{2+}$-binding loops that affect the probability of evoked release, consistent with the two-state model of *Figure 1D and E*. The finding that mutations in two basic residues that are near each other (K603 and K720) lead to opposite effects (loss- or gain-of-function) suggests that release is modulated by a delicate balance between inhibitory and stimulatory interactions involving the C$_1$C$_2$B region, which is also a key aspect of the model. Note that, while the K603E mutation strongly impaired Ca$^{2+}$-independent liposome binding and bridging, the K720E mutation impaired liposome binding to a lesser extent and still allowed liposome clustering (*Figures 2, 3B and D*, *Figure 3—figure supplement 1A*). Thus, whereas the strong impairment of membrane binding caused by the K603E mutation likely underlies the disruption of vesicle priming, the moderate impairment of binding caused by the K720E mutation may not be sufficient to impair priming but may facilitate the switch to the slanted orientation, thus enhancing evoked release (see discussion). Among the basic residues near the C$_2$B domain Ca$^{2+}$-binding sites, R769 appears to have a similar importance for Ca$^{2+}$-independent membrane binding and priming as K603, whereas, as expected, K706 is not involved in priming but plays an important function in controlling release probability. The physiological effects of the mutations are recapitulated at least to some extent in our liposome fusion assays, as the K603E and R769E mutations that impair priming (*Figure 6A*) lead to the strongest disruption of Ca$^{2+}$-independent fusion among the single mutations (*Figure 5C, D and F*), and the K706E/R769E double mutation but not the K603E/K720E mutation impaired Ca$^{2+}$-dependent fusion (*Figure 5A, B and E*), in correlation with the stronger disruption of Ca$^{2+}$-evoked release caused by K706E/R769E (*Figure 7B*).

## Distinct effects of mutations in the C$_1$C$_2$B region of Munc13-1 on short-term plasticity

We next investigated how the mutations in basic residues of the C$_1$C$_2$B region affect synaptic responses during repetitive stimulation by applying 10 Hz stimulus trains on autaptic cultures of Munc13-1 DKO neurons rescued with WT and comparing them side-by-side with rescues by the Munc13-1 mutants. Analysis of the data is complicated because changes in EPSC amplitudes during repetitive stimulation can be caused by various mechanisms, including depletion of the RRP, alterations in the efficiency of replenishment of the RRP and changes in the release probability. To help distinguish between these mechanisms, we prepared plots of normalized responses, which inform on differences in the extent of depression or facilitation during the stimulus trains, and plots of absolute EPSC amplitudes, which can help clarify the mechanisms related to use and re-fill of primed vesicles (*Shin et al., 2010*; *Figure 8*, *Figure 8—source data 1*). Nevertheless, interpretation of some of the data was not straightforward, and there may be alternative explanations to those offered below, which are based in part on the equilibrium proposed in *Figure 1D,E*.

As expected (*Rosenmund et al., 2002*), DKO neurons rescued with WT Munc13-1 displayed substantial depression in the beginning of the stimulus train that can be attributed to RRP depletion. The R769E single mutant exhibited stronger depression than WT Munc13-1 (*Figure 8L*), which likely arises from impairment in the kinetics of RRP replenishment, as this mutant had a strong defect in priming but no deficit in Pvr (*Figure 6*). However, the K603E mutant also had impaired priming and yet it exhibited similar depression to WT Munc13-1 in the beginning of the train, with a tendency to have higher EPSC amplitudes than WT as the train progressed (*Figure 8C*). These findings suggest that the

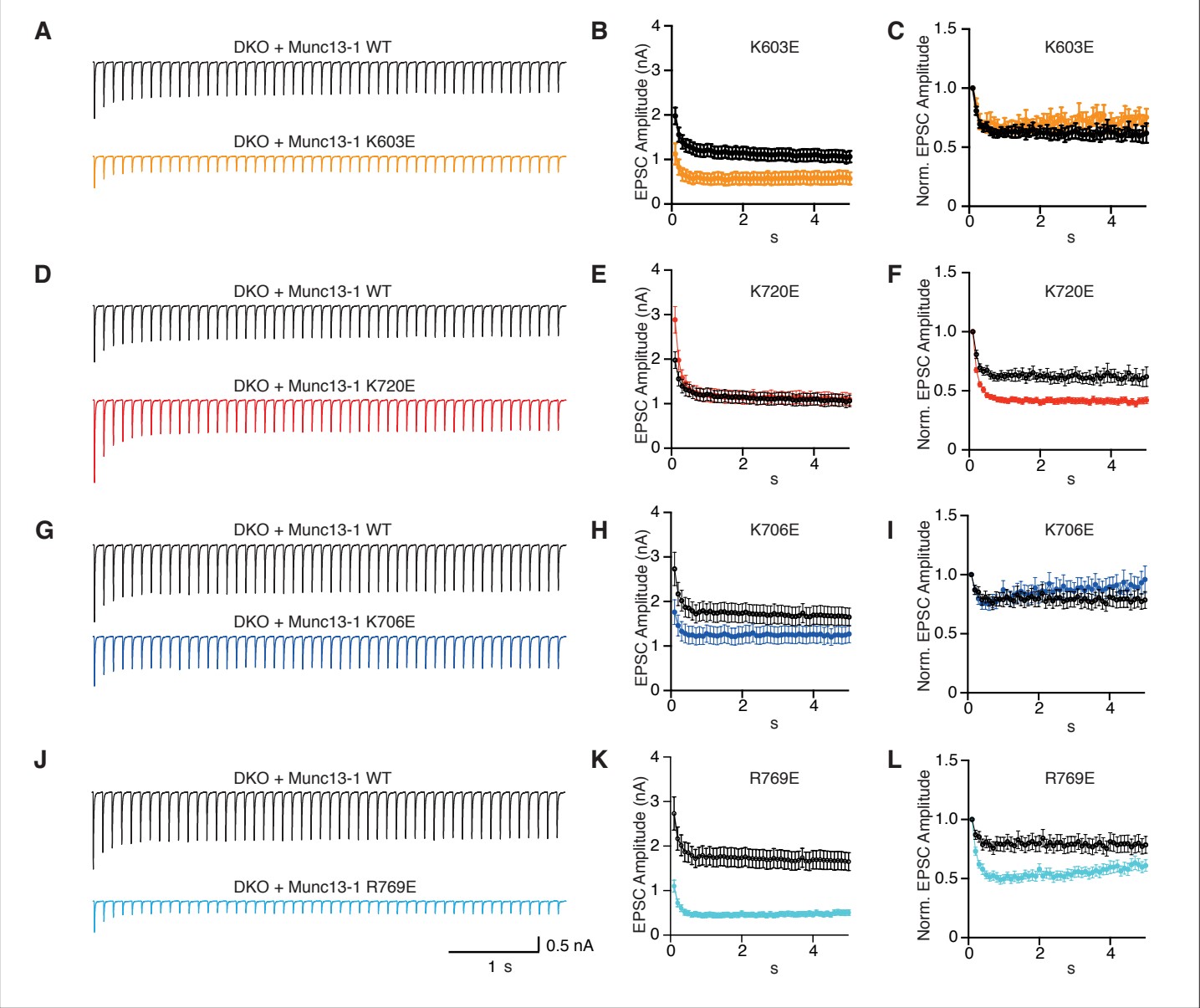

**Figure 8.** Short-term plasticity behavior of single point mutations within the Munc13-1 $C_1$-$C_2$B region. (**A**) Average EPSC traces recorded during a 10 Hz train of 50 action potentials and summary graphs of absolute (**B**) and normalized (**C**) EPSC amplitudes of autaptic hippocampal Munc13-1/2 DKO neurons expressing Munc13-1 WT (black) n = 71/5 and Munc13-1 K603E (orange) n = 67/5. Average 10 Hz train (**D**) and summary graphs of absolute (**E**) and normalized (**F**) EPSC amplitudes of autaptic hippocampal Munc13-1/2 DKO neurons rescued with Munc13-1 WT (black) n = 71/5 and Munc13-1 K720E (red) n = 67/5. Average 10 Hz train (**G**) and summary graphs of absolute (**H**) and normalized (**I**) EPSC amplitudes of autaptic hippocampal Munc13-1/2 DKO neurons rescued with Munc13-1 K706E (blue) n = 47/4. Average 10 Hz train (**J**) and summary graphs of absolute (**K**) and normalized (**L**) EPSC amplitudes of autaptic hippocampal Munc13-1/2 DKO neurons rescued with Munc13-1 WT (black) n = 52/4 and Munc13-1 R769E (light blue) n = 46/4. Action potentials artifacts were blanked.

The online version of this article includes the following figure supplement(s) for figure 8:

**Source data 1.** Average EPSC amplitudes and normalized EPSC amplitudes in *Figure 8*.

fusogenicity of primed vesicles is altered during the stimulus train and is consistent with the notion that, as intracellular $Ca^{2+}$ accumulates during repetitive stimulation, $Ca^{2+}$-binding to the Munc13-1 $C_2$B domain facilitates release (*Shin et al., 2010*) because it favors more slanted orientations of Munc13-1 (*Figure 1E*) that mediate release more efficiently than perpendicular orientations. Thus, based on this model, K603 contributes to binding of the $C_1C_2$B region to the plasma membrane in the orientations present in the absence of $Ca^{2+}$ that mediate initial priming, but not in the slanted orientations favored

upon Ca$^{2+}$ binding to the C$_2$B domain that are increasingly populated and lead to enhanced fusogenicity of the primed vesicles during the stimulus train (*Figure 1B and C*, *Figure 1—figure supplement 2*). In contrast, R769 participates in membrane binding in both orientations, consistent with the stronger depression caused by the R769E mutation compared to the K603E mutation.

The K720E mutant displayed stronger depression than WT, based on normalized EPSC plots (*Figure 8F*), but plots of absolute EPSC amplitudes show that the K720E mutant starts with higher amplitudes that depress faster over time, reaching the same steady state as WT (*Figure 8E*). Thus, the stronger depression observed initially for the K720E mutant is likely due to the initially higher release probability exhibited by this mutant (*Figure 6C*) but, based on the two-state model of *Figure 1D and E*, the K720E mutation does not affect release later during the stimulus training because K720 does not participate in binding in the slanted orientations favored by Ca$^{2+}$ accumulation. In contrast, the K706E mutation led to depression that was similar to that observed for WT Munc13-1 in normalized EPSC plots (*Figure 8I*), and absolute EPSC amplitudes show that the EPSCs observed for the K706E mutation were smaller than those of WT throughout the stimulus train (*Figure 8H*). Since the Pvr of the K706E mutant was lower than that of WT (*Figure 6C*), these observations suggest that the release probability remained low for the K706E mutant throughout the stimulus train. The effects

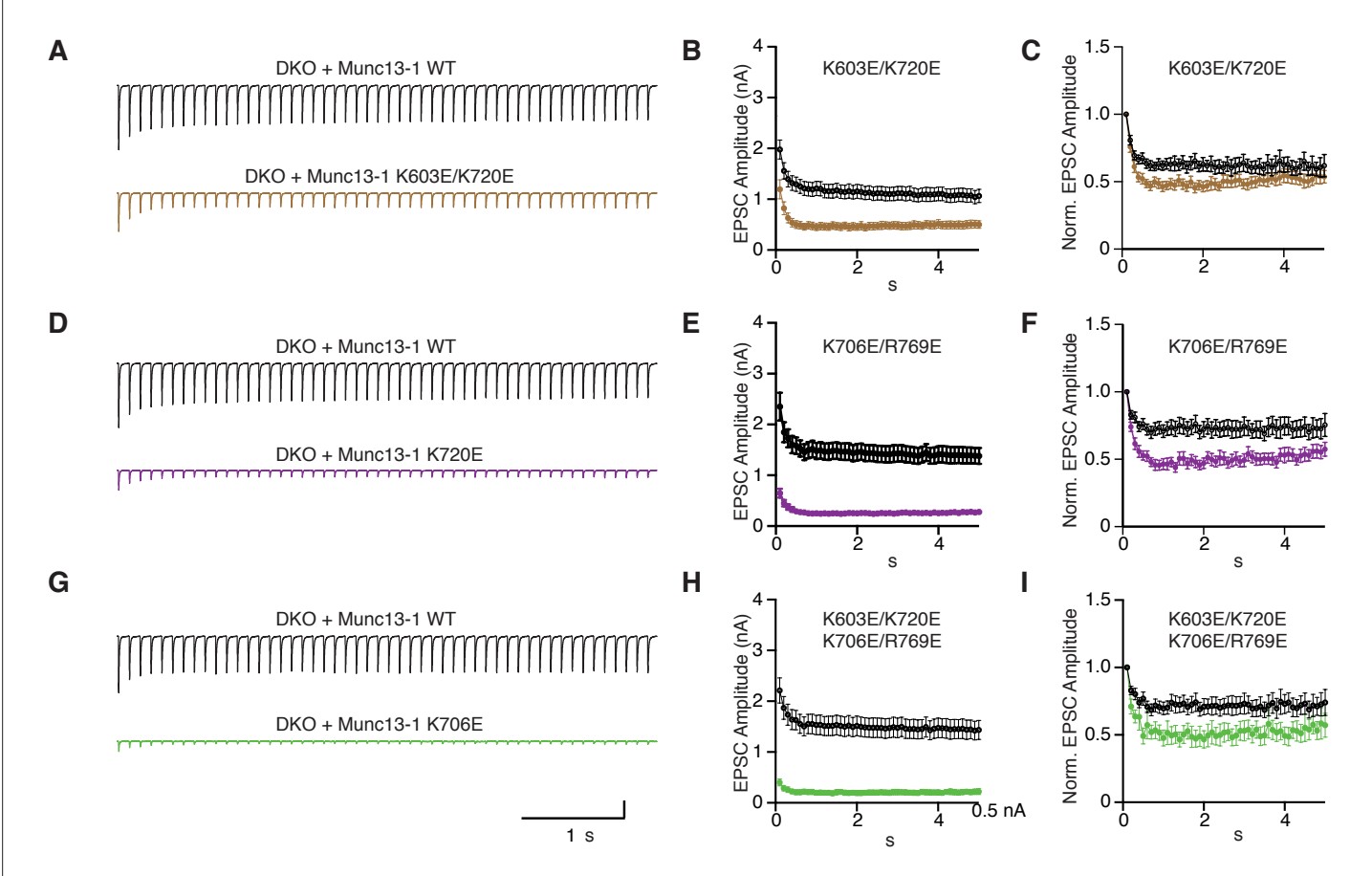

**Figure 9.** Short term plasticity behavior of combined point mutations within the Munc13-1 C$_1$-C$_2$B region. (**A**) Average EPSC traces recorded during a 10 Hz train of 50 action potentials and summary graphs of absolute (**B**) and normalized (**C**) EPSC amplitudes from autaptic hippocampal glutamatergic Munc13-1/2 DKO neurons expressing Munc13-1 WT (black) n = 71/5 and Munc13-1 K603E/K720E (brown) n = 61/5. Average EPSC traces recorded during a 10 Hz train (**D**) and summary graphs of absolute (**E**) and normalized (**F**) EPSC amplitudes from autaptic hippocampal glutamatergic Munc13-1/2 DKO neurons expressing Munc13-1 WT (black) n = 77/6 and Munc13-1 K706E/R769E (purple) n = 48/6. Average 10 Hz train (**G**) and summary graphs of absolute (**H**) and normalized (**I**) EPSC amplitudes from autaptic hippocampal glutamatergic Munc13-1/2 DKO neurons expressing Munc13-1 WT (black) n = 65/5 and Munc13-1 K603E/K720E/K706E/R769E (green) n = 17/5. Action potential artifacts were blanked.

The online version of this article includes the following figure supplement(s) for figure 9:

**Source data 1.** Average EPSC amplitudes and normalized EPSC amplitudes in *Figure 9*.

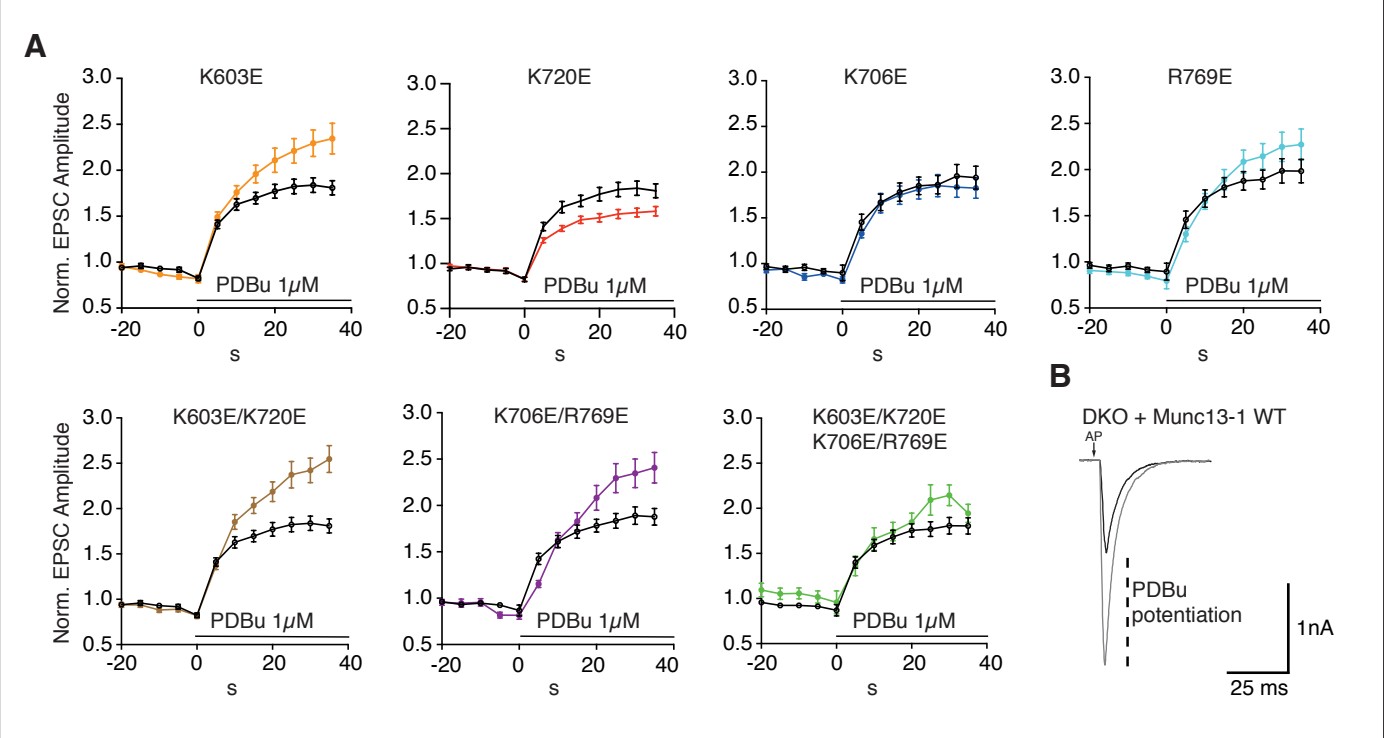

**Figure 10.** Analysis of neurotransmitter release potentiation induced by phorbol ester-activation of Munc13-1 $C_1$ domain. (**A**) Summary graphs of potentiation of EPSC amplitudes by 1 µM PDBu evoked at 0.2 Hz. EPSCs were normalized to the first EPSC amplitude recorded in extracellular solution from Munc13-1/2 DKO neurons expressing Munc13-1 WT (black) or Munc13-1 $C_1$-$C_2$B mutants (color code). Munc13-1 K603E (orange) n = 56/5 (WT n = 58/5), Munc13-1 K720E (red) n = 57/5 (WT n = 58/5), Munc13-1 K706E (blue) n = 40/4 (WT n = 38/4), Munc13-1 R769E (light blue) n = 38/4 (WT n = 36/4), Munc13-1 K603E/K720E (brown) n = 56/5 (WT n = 58/5) Munc13-1 K706E/R769E (purple) n = 42/6 (WT n = 57/6) and Munc13-1 K603E/K720E/K706E/R769E (green) n = 27/5 (WT n = 54/5). Solid symbols in the graph represent average normalized EPSC amplitudes ± SEM at each time point. (**B**) Exemplary EPSC traces from Munc13-1/2 DKO neurons expressing Munc13-1 WT before (black) and at 30 s of PDBu application (gray).

The online version of this article includes the following figure supplement(s) for figure 10:

**Source data 1.** Normalized EPSC amplitudes in *Figure 10*.

observed for the K706E mutation are opposite to those caused by a mutation in the corresponding lysine residue of ubMunc13-2 to tryptophan (*Shin et al., 2010*) and may arise because the mutation hinders the transition to the slanted orientation, but they may also reflect a decreased fusogenicity of primed vesicles. Finally, the effects observed for the K603E/K720E and K706E/R769E double mutants (*Figure 9A–F*, *Figure 9—source data 1*) and the K603E/K720E/K706E/R769E quadruple mutants (*Figure 9G–I*) appeared to be a combination of the effects caused by the single mutations, with some dominance by the K603E and the R769E mutations.

At synapses, phorbol esters increase the release probability at least in part by activating Munc13-1 through its $C_1$ domain (*Basu et al., 2007*; *Rhee et al., 2002*), mimicking the effects of DAG. The two-state model predicts that this potentiation arises because DAG/phorbol esters favor the slanted orientations of Munc13-1 that facilitate SNARE complex formation and fusion. To test this notion, we analyzed the effects of the mutations in basic residues of the Munc13-1 $C_1C_2$B region on potentiation by a phorbol ester [phorbol 13,14-dibutyrate (PDBu)]. PDBu caused a robust potentiation of EPSCs in DKO autaptic cultures rescued with WT Munc13-1 (*Figure 10A and B*, *Figure 10—source data 1*), as expected, and the K603E and R769E mutants exhibited even higher potentiation (*Figure 10A*). These results suggest that PDBu partially compensates for the priming defects caused by the K603E and R769E mutations. In contrast, the K720E mutant displayed less potentiation than WT Munc13-1, which, based on the two-state model, may arise because the K720E mutation already facilitates the transition from the perpendicular to the slanted orientation and thus has an intrinsically higher release probability. Surprisingly, the potentiation observed for the K706E mutant was similar to that observed for WT Munc13-1. Although this observation might suggest that this mutation does not affect the

transition between the LS and TS states, it is also plausible that the mutation does alter the transition but in addition affects downstream events that lead to synaptic vesicle fusion (see discussion). The K603E/K720E and K706E/R769E double mutants exhibited similar enhancements of PDBu-induced potentiation as the K603E and R769E single mutants, respectively, showing again that these mutations dominate the phenotypes of the double mutants. The results obtained for the rescue with the K603E/K720E/K706E/R769 quadruple mutant need to be interpreted with caution because this mutant exhibited a high-number of non-responsive neurons and the EPSCs for the responsive neurons were very weak. However, it is noteworthy that we still were able to observe PDBu potentiation for this mutant.

Overall, these results show that basic residues in the Munc13-1 $C_1$-$C_2B$ region influence the potentiation of synaptic responses by PDBu and, together with the data obtained with repetitive stimulation, they support the notion that two faces of the $C_1$-$C_2B$ region are involved in Munc13-1-dependent short-term plasticity.

## Discussion

Research for over three decades has led to major advances in our understanding of the basic mechanisms underlying the priming and fusion of synaptic vesicles to release neurotransmitters, including the notions that Munc18-1 and Munc13-1 coordinate assembly of SNARE complexes in an NSF-αSNAP-resistant manner (*Ma et al., 2013*; *Prinslow et al., 2019*) and that this pathway enables the multiple modes of regulation of release probability during presynaptic plasticity that depend on Munc13-1 (*Park et al., 2017*; *Sitarska et al., 2017*; *Stepien et al., 2019*). This large multiple domain protein plays fundamental roles in this process by accelerating opening of the syntaxin-1 closed conformation (*Ma et al., 2011*; *Yang et al., 2015*) and bridging the vesicle and plasma membranes (*Liu et al., 2016*; *Quade et al., 2019*). Thus, it seems likely that Munc13-1-dependent regulation of neurotransmitter release involves at least in part alteration of these activities. Indeed, the finding that the $C_1C_2B$ region has two putative faces that can bind to membranes led to an intriguing model postulating that membrane bridging by Munc13-1 in two orientations modulates neurotransmitter release (*Xu et al., 2017*). Here, we have tested this model and examined the role of the $C_1C_2B$ region in neurotransmitter release. We find that basic residues in this region play key roles in SV priming and in dictating the vesicular release probability, and also modulate short-term plasticity. The finding that two faces of the Munc13-1 $C_1C_2B$ region play important roles in neurotransmitter release support the model of *Figure 1D and E*, but other interpretations are possible.

The notion that the highly conserved C-terminal region of Munc13-1 containing the $C_1$, $C_2B$, MUN and $C_2C$ domains bridges the vesicles and plasma membranes arose naturally from the discovery of its rod-like architecture, with membrane binding domains on opposite ends of the highly elongated MUN domain (*Liu et al., 2016*; *Xu et al., 2017*). At the same time, the structural studies revealed that the Munc13-1 $C_1C_2B$ region contains a large polybasic face adjacent to the region containing the DAG/phorbol ester-binding site of the $C_1$ domain and the $Ca^{2+}$/$PIP_2$-binding region of the $C_2B$ domain (*Figure 1B and C*), suggesting that Munc13-1 can bind to the plasma membrane through two different faces that yield different orientations (*Xu et al., 2017*). Although our MD simulations need to be interpreted with caution because of the limited simulation times, they yield binding modes that make sense from a biophysical point of view and indicate that membrane interactions mediated by the polybasic face involve multiple ionic interactions in an extensive surface, leading to an almost perpendicular orientation of the Munc13-1 rod with respect to the membrane. In contrast, the simulations suggest that interactions mediated by the DAG/$Ca^{2+}$/$PIP_2$-binding face involve a smaller area and are favored by $Ca^{2+}$-phospholipid coordination and insertion of hydrophobic residues into the membrane in addition to ionic interactions, leading to a very slanted orientation (*Figure 1D and E*, *Figure 1—figure supplements 1 and 2*). The relevance of the two binding modes observed in the simulations is supported by the biochemical and physiological effects of the mutations described here and those caused by mutations that disrupt DAG/phorbol ester binding to the $C_1$ domain (*Rhee et al., 2002*) or $Ca^{2+}$ binding to the $C_2B$ domain (*Shin et al., 2010*). We also note that the structure of the $C_1C_2BMUN$ fragment appears to be rather rigid (*Xu et al., 2017*) and there is no apparent reason why $Ca^{2+}$ binding to the $C_2B$ domain should alter the overall structure of the Munc13-1 C-terminal region. Hence, although our data do not demonstrate that two orientations of Munc13-1 with respect to the plasma membrane are physiologically relevant, this notion is supported by the functional importance of the two binding faces, as they are expected to lead to two drastically different orientations.

It is important to realize that the orientation of Munc13-1 is likely dynamic, particularly in the presence of other forces. For instance, the perpendicular orientation of Munc13-1 may facilitate initiation of SNARE core complex assembly but prevent full assembly, resulting in a tug-of-war between Munc13-1 and the SNAREs in the absence of $Ca^{2+}$ that is dramatically tilted toward full SNARE complex assembly and fusion when $Ca^{2+}$ binding to the $C_2B$ domain favors slanted orientations. However, the balance can also be tilted towards slanted orientations in the absence of $Ca^{2+}$ by the energy associated with further assembly of the SNARE complex and by proteins that bind to the SNARE complex such as Syt1 and complexins (see below). Indeed, a large amount of physiological data can be explained with a model postulating that primed vesicles exist in a dynamic equilibrium between a loosely primed state (LS), in which SNARE complexes are partially assembled and Munc13-1 bridges the vesicle and plasma membrane in a perpendicular orientation, and a tightly primed state (TS) where the SNARE complex is more fully assembled and Munc13-1 bridges the membranes in a slanted orientation (*Neher and Brose, 2018*). In this scenario, vesicles in the TS state are much more likely to be released by an action potential than those in the LS state, and accumulation of $Ca^{2+}$ during repetitive stimulation increases the vesicular release probability because it shifts the equilibrium toward the TS state. Note, however, that methods used to measure the RRP, such as application of sucrose, likely release vesicles in both the LS and TS states. This two-state model and the proposed nature of the two membrane-binding modes of the $C_1$-$C_2B$ region underlying the two states are supported by our data in combination with previous studies (*Rhee et al., 2002*; *Shin et al., 2010*).

The strong impairments of liposome binding and clustering in the absence of $Ca^{2+}$ caused by the K603E and R769E mutations in Munc13-1 $C_1C_2BMUNC_2C$ (*Figures 2 and 3*) show that the polybasic face is indeed critical for $Ca^{2+}$-independent membrane binding and membrane bridging. These mutations also caused the strongest impairments of $Ca^{2+}$-independent liposome fusion among the single mutants (*Figure 5C, D and F*). The finding that the K720E mutation in the polybasic face has smaller effects on $Ca^{2+}$-independent liposome binding, clustering and fusion suggests that K720 has a smaller energetic contribution to membrane binding than those of K603 and R769. The K706E mutation exhibited the smallest inhibitory effects on liposome binding and fusion in the absence of $Ca^{2+}$, which correlates with the fact that K706 is not in the polybasic face.

$Ca^{2+}$ enhanced the affinity of all the mutants for liposomes (*Figure 2*) and increased the clustering activity of the mutants that had the weakest clustering ability (K603E, R769E, and the two double mutants), except the quadruple mutant (*Figure 3*). Hence, although we did not observe $Ca^{2+}$-induced increases in liposome binding and clustering for WT $C_1C_2BMUNC_2C$ because its liposome affinity and clustering activity are very high, $Ca^{2+}$ did increase binding and clustering when these activities were impaired by mutation. These increases may arise in part because $Ca^{2+}$ binding to the $C_2B$ domain enhances the overall positive electrostatic potential and in part because the $Ca^{2+}$-dependent binding mode with slanted orientations occurs with a higher affinity than the $Ca^{2+}$-independent binding mode involving the polybasic face. The four single mutants and the K603E/K720E double mutant supported $Ca^{2+}$-dependent fusion with comparable efficiency as WT $C_1C_2BMUNC_2C$ (*Figure 5A, B and E*) even though the $Ca^{2+}$-dependent clustering activity of all these mutants was somewhat lower than that of WT (*Figure 3*), suggesting that fusion efficiency can remain very high as long as the clustering activity is above a certain threshold. Interestingly, the double K706E/R769E mutation in the $C_2B$ domain $Ca^{2+}$-binding sites disrupted $Ca^{2+}$-dependent liposome fusion severely (*Figure 5A, B and E*) even though its clustering activity is comparable to that of C1C2BMUNC2C with the K603E/K720E double mutation in the polybasic face (*Figure 3*). This finding strongly supports the notion that a $Ca^{2+}$-induced switch to a slanted orientation is critical for $Ca^{2+}$-induced enhancement of liposome fusion.

Our electrophysiological studies support the biological relevance of at least some of the results obtained in our in vitro assays and reveal additional effects of the mutations that could not be discerned with the liposome fusion assays, providing critical evidence supporting the functional relevance of the two faces of the Munc13-1 $C_1C_2B$ region. The observation that only the K603E and R769E mutants among the single Munc13-1 mutants exhibited severely impaired SV priming (*Figure 6A*) clearly correlates with the much stronger disruption of $Ca^{2+}$-independent liposome binding, clustering and fusion caused by these mutations, compared to the K706E and K720E single mutants (*Figures 2, 3, 5C, D and F*). These results provide compelling evidence that the polybasic face of the Munc13-1 $C_1C_2B$ region plays a critical function in synaptic vesicle priming. Together with the key importance of the $C_2C$ domain for membrane bridging and SV priming (*Quade et al., 2019*), these data strongly

support the notion that this critical function involves bridging of the vesicle and plasma membranes by respective interactions involving the $C_2C$ domain and the $C_1C_2B$ region of Munc13-1. The K603E and R769E mutants also exhibited similar impairments in $Ca^{2+}$-evoked release that arose because of the defects in priming, as the Pvr remained analogous to that of WT Munc13-1 (*Figure 6B–C*), and the potentiation of release by PDBu was somewhat larger for both mutants than for WT Munc13-1 (*Figure 10A*). These observations suggest that binding of PDBu to the $C_1$ domain can partially compensate for the defects in priming caused by these two mutations by favoring the binding mode involving the slanted orientation, which increases vesicle fusogenicity. Interestingly, we observed a considerably stronger depression during a 10 Hz stimulus train for the R769E mutant than for the K603E mutant (*Figure 8*). This finding supports the notion that accumulation of $Ca^{2+}$ during repetitive stimulation facilitates the transition to slanted orientations, which are expected to be destabilized by the R769E mutation but not by the K603E mutation.

It was surprising that the K720E mutation in the polybasic interface did not impair SV priming and enhanced $Ca^{2+}$-evoked release as well as the Pvr, in contrast to the effects of the K603E in a nearby basic residue (*Figure 6A–C*). However, as mentioned above, there is likely a delicate balance between inhibitory and stimulatory interactions within the release machinery, and some of the interactions involving the polybasic region that are important for priming initiated by a perpendicular orientation of Munc13-1 need to be released to transit to slanted orientations that support full zippering of the SNARE complex and membrane fusion. Since the K720E mutation has only a modest effect on $Ca^{2+}$-independent liposome binding and clustering (*Figures 2 and 3D*), a likely explanation of our results is that the mutant retains sufficient membrane affinity to fully support priming and facilitates the transition to the more active slanted orientations, yielding a higher release probability. This model is consistent with the finding that, in contrast to the K603E mutant, the K720E mutant exhibited less potentiation of release by PDBu than WT (*Figure 10A*), likely because this mutation already promotes the slanted orientations that are favored by PDBu binding to the $C_1$ domain. In 10 Hz stimulus trains, the K720E mutant depressed initially more than WT Munc13-1, but later exhibited analogous EPSC amplitudes as WT (*Figure 8D–F*), likely because accumulation of $Ca^{2+}$ during repetitive stimulation already induces slanted orientations that do not involve interactions of K720 with the membrane. Nevertheless, we cannot rule out the possibility that the effects of the K720E mutation arise from a different mechanism, for instance if the mutation alters an as yet unidentified interaction of Munc13-1.

The K706E mutation did not affect SV priming, in agreement with the notion that K706 does not participate in the interactions of the polybasic interface that mediate priming, but this mutant did decrease $Ca^{2+}$-evoked release and hence the probability of release (*Figure 6A–C*). These effects are opposite to those caused by mutation of the corresponding lysine residue of ubMunc13-2 to Trp (the KW mutant) (*Shin et al., 2010*), which caused a gain-of-function, and could arise in principle because the K706E mutation hinders the transition to slanted orientations. This interpretation is consistent with the observation that EPSCs remained lower than WT for the K706E mutant during 10 Hz repetitive stimulation (*Figure 8G–I*). However, the observation that PDBu potentiation was similar for this mutant and WT Munc13-1 (*Figure 10A*) suggests that the effects of the K706E mutation might not be related to the transition to slanted orientations but rather to another mechanism that directly influences fusion. For instance, the Munc13-1 $C_2B$ domain might cause membrane perturbations analogous to those that are believed to underlie the function of the Syt1 $C_2$ domains in triggering release (*Fernández-Chacón et al., 2001*; *Rhee et al., 2005*). It is also possible that the phenotypes caused by the K706E mutation and other mutations studied here reflect effects of Munc13-1 in more than one step leading to release, which complicates the interpretation of the data.

Despite these uncertainties in the interpretation of our results, the overall data strongly support the notions that the Munc13-1 $C_1C_2B$ region plays a critical role in SV priming and that this region underlies two types of interactions with the plasma membrane that yield different orientations of Munc13-1. This model is also consistent with the previous observation that a mutation that is expected to unfold the Munc13-1 $C_1$ domain leads to decreased priming but increased release probability in mice (*Basu et al., 2007*; *Rhee et al., 2002*), as the absence of the $C_1$ domain is expected to impair the initial binding to the plasma membrane that is important for priming but likely helps to adopt slanted orientations that facilitate release. Similarly, the finding that deletion of the $C_1$ domain or the $C_2B$ domain of *C. elegans* unc-13 enhances release but deletion of both domains strongly impairs release (*Michelassi et al., 2017*) suggest that both the $C_1$ domain and the $C_2B$ domain are important to stabilize the

perpendicular orientations that mediate priming but hinder transition to the active, slanted orientations; however, at least one of the two domains needs to be present to mediate the interaction with the plasma membrane.

Further research will be necessary to better understand the nature of the LS and TS states and the factors that influence the postulated equilibrium between them. Thus, the equilibrium is expected to be affected by Syt1 and complexins, which are believed to maintain trans-SNARE complexes in a state that is ready for fast release but is inhibited to prevent premature fusion (*Voleti et al., 2020*), and that most likely corresponds to the TS state. Note that the release probability and the RRP are reduced in Syt1 KO mice (*Chang et al., 2018*) while, in the absence of complexins, the release probability is also decreased but not the RRP (*Reim et al., 2001*). Since Syt1 promotes trans-SNARE complex assembly and complexins protect trans-SNARE complexes against disassembly by NSF/αSNAP (*Prinslow et al., 2019*), stabilizing the C-terminus of the SNARE four-helix bundle (*Chen et al., 2002*), it is tempting to speculate that Syt1 facilitates initial formation of LS and complexins shift the equilibrium toward TS. Other proteins such as CAPS, which has functions related to those of Munc13s (*Jockusch et al., 2007*), may also affect the equilibrium between the two primed states. Clearly, much we have learned but much remains to be learned about this fascinating system.

## Materials and methods
### Molecular dynamics simulations

A square lipid bilayer of 19.347 nm x 19.347 nm was built in the membrane builder module in the Charm-gui (*Jo et al., 2008*) website (https://charmm-gui.org/). The bilayer had the following composition, which mimics that of the plasma membrane (*Chan et al., 2012*), except for having somewhat higher amounts of $PIP_2$ and DAG (the number of molecules of each lipid is indicated; percent is indicated in parenthesis): upper leaflet, 315 (45%) cholesterol, 56 (8%) 16:0-18:1 phosphatidylcholine (POPC), 91 (13%) 18:0-22:6 phosphatidyltethanolamine, 42 (6%) 18:0-22:4 phosphatidyltethanolamine (SAPE), 70 (10%) 18:0-18:1 phosphatidylserine (SOPS), 70 (10%) 18:0-22:6 phospatidylserine (SDPS), 35 (5%) 18:0-20:4 phosphatidylinositol 4,5-bisphosphate (SAPI2D), 21 (3%) 18:0-20:4 glycerol (SAGL); lower leaflet, 315 (46.1%) cholesterol, 278 (40.6%) POPC, 42 (6.1%) SDPE, 28 (4.1%) SAPE, 21 (3.1%) SAGL. The crystal structure of the Munc13-1 $C_1C_2$BMUN fragment lacking residues 1,408–1,452 (which correspond to a long variable loop) (PDB code 5UE8) lacks multiple loops for which no density could be observed (*Xu et al., 2017*). To build a complete model of this fragment (except for residues 1,408–1,452), the coordinates of this crystal structure were merged with those of the missing loops in the NMR structure of the $C_1$ domain (PDB code 1Y8F) (*Shen et al., 2005*), the crystal structure of the $Ca^{2+}$-bound $C_2$B domain (PDB code 6NYT) (*Shin et al., 2010*) and the refined crystal structure of the MUN domain (PDB code 5UF7) (*Xu et al., 2017*), after superimposing the common coordinates of these structures. Two additional missing loops were modeled with Robetta (*Song et al., 2013*) (https://robetta.bakerlab.org/). We placed the $Ca^{2+}$-free $C_1C_2$BMUN model above the upper leaflet in an approximately perpendicular orientation with the polybasic face close to the membrane for the $Ca^{2+}$-free simulation (black wire diagram in *Figure 1—figure supplement 1A*). For the $Ca^{2+}$-bound simulation, we built another system that included two $Ca^{2+}$ ions bound to the corresponding binding sites of the $C_2$B domain (*Shin et al., 2010*) and where $C_1C_2$BMUN was placed in a more slanted orientation such that the membrane was close to the DAG- and $Ca^{2+}/PIP_2$-binding sites of the $C_1$ and $C_2$B domains, respectively (black wire diagram in *Figure 1—figure supplement 1C*).

All computations with these systems were performed with Gromacs (*Pronk et al., 2013*; *Van Der Spoel et al., 2005*) at the BioHPC supercomputing facility of UT Southwestern or at the Texas Advanced Computing Center using the CHARMM36 force field (*Best et al., 2012*). TIP3P explicit water boxes were built for both systems (24.3 × 26.7 x 24.3 $nm^3$ for the $Ca^{2+}$-free system; 24.3 × 28.3 x 24.3 $nm^3$ for the $Ca^{2+}$-bound system), and potassium and chloride ions were added to yield a KL concentration of 145 mM and overall charge neutrality, resulting in systems of 1.55 ($Ca^{2+}$ free) and 1.64 ($Ca^{2+}$-bound) million atoms. The systems were energy minimized, heated to 310 K over the course of 1 ns in the NVT ensemble and equilibrated for one ns in the NPT ensemble. NPT production simulations were ran for 100 ns ($Ca^{2+}$-free system) or 86 ns ($Ca^{2+}$-bound system) using two fs steps and a 1.1 nm cutoff for non-bonding interactions, and periodic boundary conditions were imposed

with Particle Mesh Ewald (PME) (*Darden et al., 1993*) summation for long-range electrostatics. Pymol (Schrödinger, LLC) was used for molecular graphics.

## Munc13-1/2 double knock-out

The approved mouse gene symbols for Munc13-1 and Munc13-2 are *Unc13a* and *Unc13b*, respectively. Hence, Munc13-1/2 double knockout (DKO) mice have loss of function in alleles *Unc13a* and *Unc13b*. Munc13-1/2 DKO embryos (*Varoqueaux et al., 2002*) on an FVB/N background were used for the generation of hippocampal neuronal cultures. Munc13-1/2 DKO embryos were obtained by caesarean section of pregnant females from timed mating of *Unc13a+/-/Unc13b-/-* mice. Animal Welfare Committee of Charité – Universitätsmedizin Berlin and the Berlin state government agency for Health and Social Services approved all protocols for animal maintenance and experiments (license no. G106/20). Gender of animals used for the neuronal cultures were not distinguished.

## Hippocampal neuronal cultures

Single *Munc13-1/2* DKO hippocampal neurons seeded onto micro-islands of astrocytes (autaptic hippocampal neurons) were prepared as previously reported (*Bekkers and Stevens, 1991*). Briefly, HCl-cleaned 30 mm diameter glass coverslips were coated with agarose type II-A (Sigma-Aldrich) and dried for more that 48 hr. Pre-coated coverslips were printed by stamping (0.2 mm spot diameter and 0.5 mm spot interspace) with a cell-attachment mixture of rat tail collagen (BD Biosciences), poly-D-lysine (Sigma-Aldrich) and acetic acid (Sigma-Aldrich). Astrocytes were prepared from cerebral cortices of C57BL/6 N mice of either sex at postnatal day 0–1 (P0-1). Cortices were isolated and digested with 0.25 % trypsin-EDTA solution (Gibco) for 15 min at 37 °C. Enzymatic solution was discarded, and cortices were washed twice with Dulbecco's Modified Eagle Medium (DMEM) (Gibco) supplemented with 10 % fetal bovine serum (FBS) (PAA) and 50 IU/ml penicillin and 50 µg/ml streptomycin (Gibco). After washing, tissue was homogenized in the same medium and cell suspension was cultured in 75 cm$^2$ flasks containing DMEM supplemented with FBS and penicillin/streptomycin. After 15 days in vitro (DIV) non astrocytic cells were removed, and astrocytes were trypsinized and plated onto the microdot printed coverslips at a density of 5,000 cell/cm$^2$. Neurons were prepared from hippocampi of *Munc13-1/2* DKO embryos of either sex at embryonic day of 18.5 (E18.5). Hippocampi were isolated and digested for 45 min at 37 °C in DMEM containing 25 U/ml papain (Worthington), 1.65 mM L-cysteine (Sigma-Aldrich), 1 mM CaCl$_2$ (Sigma-Aldrich), and 0.5 mM EDTA (Merck). Digestion was stopped replacing the enzymatic solution by DMEM containing 10 % of FBS, 2.5 mg/ml albumin (Sigma-Aldrich) and 2.5 mg/ml trypsin inhibitor (Sigma-Aldrich). After enzymatic inactivation, hippocampi were mechanically dissociated in Neurobasal A (NBA) medium (Gibco) supplemented with 2 % B27 (Gibco), 1 % Glutamax (Gibco) and 50 IU/ml penicillin and 50 µg/ml streptomycin. Neurons were seeded at low density of 3,000 cells/well onto coverslips with the astrocytic micro-island placed in six well plates filled with NBA medium supplemented with B27, Glutamax and penicillin/streptomycin. Autaptic cultures were kept in an incubator at 37 °C with 5 % CO$_2$ for 15 days before they were used for electrophysiological experiments.

## Munc13-1 lentiviral rescue constructs

*Munc13-1* control and point mutants were generated with a FLAG-tagged at the C-terminus. *Munc13-1* cDNA was constructed from rat *Unc13a* splice variant by PCR amplification (*Camacho et al., 2017*). The reverse primer harbors a 3xFLAG sequence (Sigma-Aldrich) to allow the expression analysis. The corresponding *Munc13-1-FLAG* PCR product was fused to a cleavage P2A (*Kim et al., 2011*) upstream of a nuclear localized GFP sequence (*NLS-GFP*) for identification of infected neurons. Munc13-1 C1 single point mutant (Munc13-1 K603E), Munc13-1 C2B single and double point mutants (Munc13-1 K706E, Munc13-1 K720E, Munc13-1 R769E and Munc13-1 K720E/R769E) and Munc13-1 C1-C2B combined point mutants (Munc13-1 K603E/K706E, and Munc13-1 K603E/K706E/ K720E/R769E), were generated in the *NLS-GFP/P2A/Munc13-1-FLAG* by PCR using primers which contained the mutations. All Munc13-1 rescue constructs engineered were confirmed by sequencing. The DNA cassette encoding *NLS-GFP/P2A/Munc13-1-FLAG* control or mutant variants were subsequently cloned into a FUWG lentiviral shuttle vector under the control of the human *synapsin-1*. Lentiviral particles were produced as described previously (*Lois et al., 2002*) with slight modifications. Briefly, HEK293T cells were co-transfected using polyethylenimine (PEI) with FUGW shuttle vector and

two packaging plasmids pCMVd8.9 and pVSV-G. Cells were incubated for 72 hr at 32 °C and 5 % $CO_2$ in NBA medium supplemented with B27 and penicillin/streptomycin. NBA medium containing lentiviruses were harvested, passed through 0.45 µm filter, concentrated by centrifugation using an Amicon tube (Ultra-15, Ultracel-100 kDa) and stored at –80 °C. Neurons were infected 24 hr after plating with 50 µl of the different concentrated lentiviral rescue constructs per 35 mm diameter well.

## Electrophysiology on autaptic hippocampal neuronal cultures

Whole-cell voltage-clamp recordings were made from excitatory hippocampal autaptic neurons at room temperature after 15–18 days in vitro. Glass coverslip containing the autaptic neuronal cultures were immersed in extracellular saline solution consisting of (in mM): 140 NaCl, 2.4 KCl, 10 HEPES, 10 glucose, 2 $CaCl_2$ and 4 $MgCl_2$ (300 mOsm pH, 7.4). Borosilicate patch-pipette with resistances from 3 to 4 MΩ were filled with a KCl-based intracellular solution containing (in mM): 136 KCl, 17.8 HEPES, 1 EGTA, 4.6 $MgCl_2$, 2 mM $Na_2ATP$, 0.3 $Na_2GTP$, 12 creatine phosphate, and 50 U/ml phosphocreatine kinase (300 mOsm, pH 7.4). Only single autaptic neurons with a nuclear localized GFP signal were used for recording. Neurons were voltage clamped at –70 mV using a Multiclamp 700B amplifier and signals were digitized using a Digidata 1,440 A digitizer (Molecular Devices). Series resistance was compensated at 70 %. Only neurons with access resistance of <10 MΩ were used. Glutamatergic neurons were identified by their characteristic EPSC time course and by sensitivity of EPSC to application of glutamate receptor antagonist kynurenic acid (Tocris). Synaptic currents were acquired at 10 kHz using pClamp 10.2 software (Molecular Devices) and filtered at 3 kHz. Miniature excitatory postsynaptic currents (mEPSCs) were recorded for periods of at least 30 s in extracellular solution. Recordings in 3 mM Kynurenic acid were used to subtract false-positive events. Miniature events were detected using a template-based mEPSC algorithm implemented in Axograph X (AxoGraph Scientific). Excitatory postsynaptic currents (EPSCs) were induced by a 2 ms somatic depolarizing from –70–0 mV at a frequency of 0.2 Hz. The ready releasable pool (RRP) was determined by integrating the transient postsynaptic current induced by a 5 s application of 0.5 M hypertonic sucrose solution. *Pvr* was estimated as the ratio of the EPSC charge to the RRP charge. Short term plasticity was evaluated by a Pair-pulse protocol which two consecutives APs at 40 Hz or by a train of 50 APs at 10 Hz. Paired-pulse ratios were calculated by dividing the amplitude of the second EPSC to the first. PDBu induced potentiation was calculated by comparing the EPSC amplitude after PDBu application with the preceding EPSC amplitudes in extracellular solution.

## Immunocytochemistry

*Munc13-1/2* DKO hippocampal mass culture neurons were plated at a density of 25,000 cells, infected with the Munc13-1 $C_1$-$C_2B$ polybasic face mutants and fixed after 15 DIV with 4 % paraformaldehyde. After fixation, neurons were permeabilized, blocked and incubated overnight with rabbit polyclonal antibody against Munc13-1 (1:500; Synaptic System 126103), chicken polyclonal antibody against MAP2 (1:2,000; Merck Millipore AB5543) and guinea pig polyclonal antibody against VGLUT1 (1:4,000; Synaptic System, 135304). Primary antibodies were labelled with Alexa Fluor 488 AffiniPure donkey anti-guinea pig IgG, Rhodamine Red-X AffiniPure donkey anti-chicken IgG and Alexa Fluor 647 AffiniPure donkey anti-rabbit IgG (1:500; Jackson ImmunoResearch) for 1 hr at RT. Coverslips were mounted with Mowiol 4–88 antifade medium (Polysciences Europe). Fixed neurons were imaged on a confocal laser-scanning microscope Leica TCS SP8 with identical settings used for all samples. Neuronal cultures were visualized using a 63× oil immersion objective. Images were acquired using Leica Application Suite X (LAsX) software at 1,024 × 1,024 pixels resolution using a *z*-series projection of 10–12 images with 0.3 µm depth intervals. Six independent neurons per group for each cultured and two different cultures were imaged and analyzed using ImageJ software.

## Electrophysiological data analysis and statistics

Electrophysiological data were analyzed offline using Axograph X software version 1.4.3 (AxoGraph Scientific). Statistical analyses were carried out with GraphPad Prism software version 8 (GraphPad software). Number of neurons and cultures used for the statistical analyses are specified within the bar plots. Data was tested for normality by a D'Agostino-Pearson test showing a non-normal distribution. Electrophysiological data are presented as normalized means to the corresponding WT control group± standard error for means (SEM), except for the Pvr and PPR that are reported as absolute means ±

SEM. Statistical comparison between the different mutant groups was performed with Mann–Whitney $U$ test. The significance level was set at $p = 0.05$.

## Plasmids and recombinant proteins

Plasmids used to express the following proteins, as well as methods for expression and purification in bacteria were described previously: full-length *Homo sapiens* SNAP-25A with and without its four cysteines mutated to serine, full-length *Rattus norvegicus* synaptobrevin-2, full-length *Rattus norvegicus* Munc18-1 WT and D326K, full-length *Rattus* syntaxin-1A, full-length *Cricetulus griseus* NSF V155M mutant, *Rattus norvegicus* synaptotagmin-1 57–421 C74S/C75A/C77S/C79I/C82L/C277S (a kind gift from Thomas Söllner), and full-length *Bos taurus* α-SNAP in *E. coli* were described previously (*Chen et al., 2006*; *Dulubova et al., 2007*; *Dulubova et al., 1999*; *Liu et al., 2016*; *Liu et al., 2017*; *Ma et al., 2013*; *Prinslow et al., 2019*; *Sitarska et al., 2017*; *Stepien and Rizo, 2021*). A plasmid for bacterial expression of *Rattus Norvegicus* His6-Munc13-1 residues 529–1,735 lacking a large variable loop (residues 1,408–1,452) to improve solubility (*Ma et al., 2011*) was described previously (*Quade et al., 2019*). Standard PCR-based recombinant DNA techniques with custom-designed primers based on the parent DNA were used to create expression vectors for the following mutants of Munc13-1 $C_1C_2BMUNC_2C$: K603E, K706E, K720E, R769E, K603E/K720E, K706E/R769E, K603E/K706E/K720E/R769E.

Expression of His6-Munc13-1 $C_1C_2BMUNC_2C$ (WT and mutants) encoded in a pET28a vector was performed in *E. coli* BL21 (DE3) cells. Transformed cells were grown in the presence of 50 µg/ml kanamycin to an OD600 of ~0.8 and induced overnight at 16 °C with 500 µM IPTG. Cells were harvested by centrifugation and re-suspended in 50 mM Tris, pH 8, 250 mM NaCl, 1 mM TCEP, 10 % glycerol (v/v) prior to lysis. Cell lysates were centrifuged for 30 minutes at 48,000 x g to clarify the lysate and then incubated with Ni-NTA resin for 30 min at room temperature. The resin was washed with re-suspension buffer and re-suspension buffer with an additional 750 mM NaCl to remove contaminants. Nuclease treatment was performed on the beads for 1 hr at room temperature using 250 U of Pierce Universal Nuclease (Thermo Scientific) per liter of cells. Protein was eluted using re-suspension buffer with 500 mM imidazole and dialyzed against 50 mM Tris, pH 8, 250 mM NaCl, 1 mM TCEP, 2.5 mM $CaCl$, 10 % glycerol (v/v), overnight at 4 °C in the presence of thrombin. The solution was re-applied to Ni-NTA resin to remove any uncleaved protein and diluted twentyfold with 20 mM Tris, pH 8, 1 mM TCEP, 10 % glycerol (v/v). Diluted protein was subjected to anion exchange chromatography using a HiTrapQ HP column (GE Life Sciences) and eluted in 20 mM Tris, pH 8, 1 mM TCEP, 10 % glycerol (v/v) with a linear gradient from 1% to 50% of 1 M NaCl. Fractions containing protein were applied to a Superdex 200 column using 20 mM Tris, pH 8, 250 mM NaCl, 1 mM TCEP, 10 % glycerol (v/v).

## Liposome clustering assays

To prepare phospholipid vesicles, 1-palmitoyl-2-oleoyl-sn-glycero-3-phosphocholine (POPC), 1,2-dioleoyl-sn-glycero-3-phospho-L-serine (DOPS), 1-palmitoyl-2-oleoyl-sn-glycero-3-phosphoethanolamine (POPE), L-a-Phosphatidylinositol-4,5-bisphosphate (PIP2), 1-palmitoyl-2-oleoyl-*sn*-glycerol (DAG), and cholesterol dissolved in chloroform were mixed at the desired ratios and then dried under a stream of nitrogen gas. T-type liposomes contained 38 % POPC, 18 % DOPS, 20 % POPE, 2 % PIP2, 2 % DAG, and 20 % Cholesterol, and V-type liposomes contained 39 % POPC, 19 % DOPS, 22 % POPE, and 20 % Cholesterol. The dried lipids were left overnight in a vacuum chamber to remove the organic solvent. The next day the lipid films were hydrated with 25 mM HEPES, pH 7.4, 150 mM KCl, 10 % glycerol (v/v), and vortexed for 5 min followed by five freeze-thaw cycles. Large unilamellar vesicles were prepared by extruding the hydrated lipid solution through a 100 nm polycarbonate filter 31 times with an Avanti Mini-Extruder. Liposome clustering induced by Munc13 fragments was analyzed using a Wyatt Dynapro Nanostar dynamic light scattering instrument (Wyatt Technology) equipped with a temperature controlled microsampler as previously described (*Liu et al., 2016*). Particle size was measured with T-liposomes (250 µM total lipid) and V-liposomes (125 µM total lipid) diluted in 25 mM HEPES, pH 7.4, 150 mM KCl, 10 % glycerol (v/v) with 100 µM EGTA, two minutes after the specified Munc13-1 fragment (500 nM) was added to the mixture, and 3 min after the addition of 600 µM $Ca^{2+}$. All incubations were performed at room temperature.

## Liposome co-sedimentation assays

Liposome co-sedimentation assays were performed as described with some modifications (*Quade et al., 2019*; *Shin et al., 2010*). Briefly, lipid mixtures containing 38 % POPC, 18 % DOPS, 19 % POPE, 2 % PIP2, 2 % DAG, 20 % cholesterol, and 1 % Rhodamine-PE were dried under a stream of nitrogen gas and kept under vacuum overnight. The next day, the lipid film was re-suspended in buffer (25 mM Hepes, pH 7.4, 150 mM KCl, 1 mM TCEP, 500 mM sucrose), frozen and thawed five times, and then extruded through a 100 nm polycarbonate filter 31 times. Liposomes were diluted in sucrose-free buffer and spun at 160,000 x g for 30 min to pellet heavy liposomes. The supernatant was removed and liposomes were re-suspended in sucrose-free buffer. Liposomes were then pelleted at 17,000 x g and re-suspended in sucrose-free buffer two more times. The final liposome concentration was estimated based on the absorbance of Rhodamine-PE in a known liposome sample. Liposome solutions containing 2 mM lipids and 2 µM protein were incubated for 30 min at room temperature. The liposomes and bound protein were pelleted by centrifugation at 17,000 x g for 20 min. The supernatant was removed and the liposomes were re-suspended in buffer. Re-suspended samples were boiled for 5 min and analyzed by SDS-PAGE and coomassie blue staining.

## Liposome fusion assays

Liposome lipid and content mixing assays were performed as previously described (*Liu et al., 2016*; *Liu et al., 2017*; *Stepien and Rizo, 2021*). To prepare the phospholipid vesicles, POPC, DOPS, POPE, PIP2, DAG, 1,2- dipalmitoyl-*sn*-glycero-3-phosphoethanolamine-N-(7-nitro-2–1,3-benzoxadiazol-4-yl) (ammonium salt) (NBD-PE), 1,2-Dihexadecanoyl-*sn*-glycero-3-phosphoethanolamine (Marina Blue DHPE), and cholesterol in chloroform were mixed at the desired ratio and dried under a stream of nitrogen gas. T-liposomes contained 38 % POPC, 18 % DOPS, 20 % POPE, 2 % PIP2, 2 % DAG, and 20 % Cholesterol, and V-liposomes contained 39 % POPC, 19 % DOPS, 19 % POPE, 20 % Cholesterol, 1.5 % NBD PE, and 1.5 % Marina Blue DHPE. The dried lipids were left overnight in a vacuum chamber to remove the organic solvent. The next day, the lipid films were hydrated with 25 mM Hepes, pH 7.4, 150 mM KCl, 1 mM TCEP, 2 % n-Octyl-β-D- glucoside (β-OG) and 10 % glycerol (v/v) by vortexing for 5 minutes. For the experiments of *Figures 3 and 4*, rehydrated lipids for T- liposomes were mixed with protein and dye to get a final concentration of 4 mM lipid, 5 µM full- length syntaxin-1, 25 µM full-length SNAP-25 with its four cysteines mutated to serine, and 4 µM R-phycoerythrin biotin-XX conjugate (Invitrogen). Rehydrated lipids for V-liposomes were mixed with protein and dye to get a final concentration of 4 mM lipid, 8 µM full-length synaptobrevin, and 8 µM Cy5-streptavidin conjugate (Seracare Life Sciences Inc). Lipid mixtures were dialyzed 1 hr, 2 hr and overnight at 4 °C in 25 mM Hepes, pH 7.4, 150 mM KCl, 1 mM TCEP, 10 % glycerol (v/v) in the presence of Amberlyte XAD-2 beads (Sigma) to remove the detergent and promote the formation of proteoliposomes. The next day, the proteoliposomes were harvested and mixed with Histodenz (Sigma) to a final concentration of 35 %. Proteoliposome mixtures were added to a centrifuge tube with 25 % Histodenz and 25 mM Hepes, pH 7.4, 150 mM KCl, 1 mM TCEP, 10 % glycerol layered on top. The proteoliposomes were spun at 4 °C for 1.5 hr at 55,000 RPM in an SW-60 TI rotor and the top layer was collected. Concentrations of the final T-proteoliposomes were measured by the *Stewart, 1980* method. V-proteoliposome concentrations were estimated from the UV-vis absorption using a standard curve made using known quantities of liposomes containing 1.5 % NBD-PE. For the experiments of *Figure 5—figure supplement 1*, we used the same protocol except for the following modifications. We used WT SNAP-25 that was dodecylated as described (*Stepien and Rizo, 2021*) and incorporated into the T-liposomes with P:L ratio 1:800 instead of SNAP-25 with its four cysteines mutated to serine, and syntaxin-1 was incorporated into the T-liposomes with P:L ratio 1:5,000. Instead of V-liposomes, we used VSyt1-liposomes containing 40 % POPC, 6.8 % DOPS, 30.2 % POPE, 20 % Cholesterol, 1.5 % NBD PE, and 1.5 % Marina Blue DHPE, synaptobrevin (1:10,000 P:L ratio) and synaptotagmin-1 (57–421) C74S/C75A/C77S/C79I/C82L/C277S (P:L ratio 1,1,000), as described (*Stepien and Rizo, 2021*).

To perform the fusion assays, T-liposomes (250 µM total lipid) were first incubated with 1 µM Munc18-1 (WT or D326K as indicated), 0.8 µM NSF, 2 µM αSNAP, 2 mM ATP, 2.5 mM Mg2+, 5 µM streptavidin, and 100 µM EGTA for 15–25 min at 37 °C, and then were mixed with V- liposomes or VSyt1-liposomes (125 µM total lipid), 1 µM SNAP-25 (only for the experiments of *Figure 5* but not for those in *Figure 5—figure supplement 1*), and WT or mutant $C_1C_2BMUNC_2C$ (0.1 µM for the experiments of *Figure 5B,C*, or 0.5 µM for the experiments of *Figure 5C,D*). After 5 min 0.6 mM Ca$^{2+}$ was

added to stimulate fusion, and 1 % β- OG was added after 25 min to solubilize the liposomes. The fluorescence signals from Marina Blue (excitation at 370 nm, emission at 465 nm) and Cy5 (excitation at 565 nm, emission at 670 nm) were recorded to monitor lipid and content mixing, respectively. The lipid mixing data were normalized to the maximum fluorescence signal observed upon detergent addition. The content mixing data were normalized to the maximum Cy5 fluorescence observed after detergent addition in control experiments without external streptavidin.

### Measuring Ca²⁺ concentrations during liposome fusion assays

Liposome lipid mixing assays were performed basically as described above for the experiments with V- and T-liposomes, except that we used different fluorescent lipids. To prepare the phospholipid vesicles, POPC, DOPS, POPE, PIP2, DAG, and cholesterol in chloroform and 1,1'-dioctadecyl-3,3,3',3'-tetramethylindodicarbocyanine, 4-chlorobenzenesulfonate salt (DiD, Invitrogen) in DMSO were mixed at the desired ratio and dried under a stream of nitrogen gas. T-liposomes contained 38 % POPC, 18 % DOPS, 20 % POPE, 2 % PIP2, 2 % DAG, and 20 % Cholesterol, and V-liposomes contained 38.5 % POPC, 19 % DOPS, 19 % POPE, 20 % Cholesterol, and 3.5 % DiD. The dried lipids were left overnight in a vacuum chamber to remove the organic solvent. The next day the lipid films were hydrated with 25 mM Hepes, pH 7.4, 150 mM KCl, 1 mM TCEP, 2 % n-Octyl-β-D- glucoside (β-OG) and 10 % glycerol (v/v) by vortexing for 5 minutes. Rehydrated lipids for T- liposomes were mixed with protein to get a final concentration of 4 mM lipid, 5 µM full-length syntaxin-1, 25 µM full-length SNAP-25 with the four cysteines mutated to serine. Rehydrated lipids for V-liposomes were mixed with protein to get a final concentration of 4 mM lipid and 8 µM full-length synaptobrevin. Lipid mixtures were dialyzed 1 hr, 2 hr and overnight at 4 °C in 25 mM Hepes, pH 7.4, 150 mM KCl, 1 mM TCEP, 10 % glycerol (v/v) in the presence of Amberlyte XAD-2 beads (Sigma) to remove the detergent and promote the formation of proteoliposomes.

To perform the fusion assays, T-liposomes (250 µM total lipid) were first incubated with 1 µM Munc18-1 wild type, 0.8 µM NSF, 2 µM αSNAP, 2 mM ATP, 2.5 mM Mg2+, 5 µM streptavidin, 100 uM EGTA, and 0.5 µM Fluo-4, pentapotassium salt for 15–25 min at 37 °C. To initiate the reaction preincubated T- liposomes were mixed with V-liposomes (125 µM total lipid), 1 µM SNAP-25, and 0.1 µM C1C2BMUNC2C. After 5 min Ca²⁺ was added to stimulate fusion, and 1 % β-OG was added after 25 min to solubilize the liposomes. Throughout the reaction lipid mixing was monitored using DiD de-quenching with excitation at 560 nm and emission measured at 670 nm. After solubilization of samples, the Fluo-4 emission intensity was measured from 505 to 540 nm with excitation at 465 nm. To measure the maximum fluorescence of Fluo-4 in each sample, 10 mM Ca²⁺ was added to each sample and the fluorescence spectrum was acquired again. Ca²⁺ concentrations were calculated as previously described (*Grynkiewicz et al., 1985*).

## Acknowledgements

We thank Berit Söhl-Kielczynski, Bettina Brokowski, Katja Pötschke, Sabine Lenz and Heike Lerch for excellent technical support, and Milo Lin for critical reading of the manuscript. We also thank the Charité Viral core facility for virus production and characterization, and Neurocure imaging core facility at the Charité Campus Mitte for services. The authors acknowledge the Texas Advanced Computing Center (TACC) at The University of Texas at Austin for providing high performance computing resources that have contributed to the research results reported within this paper (URL: http://www.tacc.utexas.edu). This research was also used computational resources provided by the BioHPC supercomputing facility located in the Lyda Hill Department of Bioinformatics, UT Southwestern Medical Center, TX (URL: https://portal.biohpc.swmed.edu). Bradley Quade was supported by NIH Training Grant T32 GM008297. This work was supported by grant I-1304 from the Welch Foundation (to JR), by NIH Research Project Award R35 NS097333 (to JR), by the German Research Council Grant CRC 958 and Reinhart Koselleck project (to CR).

## Additional information

### Funding

| Funder | Grant reference number | Author |
|---|---|---|
| National Institute of Neurological Disorders and Stroke | R35 NS097333 | Josep Rizo |
| Welch Foundation | I-1304 | Josep Rizo |
| German Research Foundation | 958 | Christian Rosenmund |
| Deutsche Forschungsgemeinschaft | | Christian Rosenmund |

The funders had no role in study design, data collection and interpretation, or the decision to submit the work for publication.

### Author contributions

Marcial Camacho, Conceptualization, Data curation, Formal analysis, Investigation, Methodology, Writing - original draft; Bradley Quade, Data curation, Formal analysis, Investigation, Methodology, Writing - original draft; Thorsten Trimbuch, Investigation; Junjie Xu, Formal analysis, Investigation, Writing - review and editing; Levent Sari, Methodology, Writing - review and editing; Josep Rizo, Conceptualization, Formal analysis, Funding acquisition, Investigation, Methodology, Project administration, Supervision, Writing - original draft; Christian Rosenmund, Conceptualization, Data curation, Formal analysis, Funding acquisition, Investigation, Methodology, Writing - original draft

### Author ORCIDs

Marcial Camacho (iD) http://orcid.org/0000-0002-2367-1259
Bradley Quade (iD) http://orcid.org/0000-0002-5330-1355
Josep Rizo (iD) http://orcid.org/0000-0003-1773-8311
Christian Rosenmund (iD) http://orcid.org/0000-0002-3905-2444

### Ethics

Animal Welfare Committee of Charité - Universitätsmedizin Berlin and the Berlin state government agency for Health and Social Services approved all protocols for animal maintenance and experiments (license no. G106/20).

### Decision letter and Author response

Decision letter https://doi.org/10.7554/eLife.72030.sa1
Author response https://doi.org/10.7554/eLife.72030.sa2

## Additional files

### Supplementary files

• Transparent reporting form

### Data availability

Source data files are provided for all data figure panels.

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

# Appendix 1

## Appendix 1—key resources table

| Reagent type (species) or resource | Designation | Source or reference | Identifiers | Additional information |
|---|---|---|---|---|
| recombinant DNA reagent | Pet28a-His6-Syx1A | *Ma et al., 2013* | | Protein expression plasmid for *E. coli* (*Rattus Norvegicus* syntaxin-1A) |
| recombinant DNA reagent | Pet28a-His6-SNAP-25A | *Chen et al., 2006* | | Protein expression plasmid for *E. coli* (*Rattus Norvegicus* SNAP-25A C84S, C85S, C90S, C92S) |
| recombinant DNA reagent | pGEX-KT-GST-Syb | *Chen et al., 2006* | | Protein expression plasmid for *E. coli* (*Rattus Norvegicus* synaptobrevin 2) |
| recombinant DNA reagent | pGEX-KG-GST-Munc18-1 | *Dulubova et al., 1999* | | Protein expression plasmid for *E. coli* (*Rattus Norvegicus* Munc18-1) |
| recombinant DNA reagent | pGEX-KG-GST-NSF | *Ma et al., 2013* | | Protein expression plasmid for *E. coli* (*Bos Taurus* NSF V155M) |
| recombinant DNA reagent | pGEX-KG-GST- α SNAP | *Ma et al., 2013* | | Protein expression plasmid for *E. coli* (*Bos Taurus* α SNAP) |
| recombinant DNA reagent | Pet28a-His6-Munc13-1-$C_1C_2BMUNC_2C$ WT | *Quade et al., 2019* | | Protein expression plasmid for *E. coli* (*Rattus Norvegicus* Munc13-1 (529–1735, °1408–1452)) |
| recombinant DNA reagent | Pet28a-His6-Munc13-1-$C_1C_2BMUNC_2C$ K603E | This paper | | Protein expression plasmid for *E. coli* (*Rattus Norvegicus* Munc13-1 (529–1735, °1408–1452)) K603E |
| recombinant DNA reagent | Pet28a-His6-Munc13-1-$C_1C_2BMUNC_2C$ K706E | This paper | | Protein expression plasmid for *E. coli* (*Rattus Norvegicus* Munc13-1 (529–1735, °1408–1452)) K706E |
| recombinant DNA reagent | Pet28a-His6-Munc13-1-$C_1C_2BMUNC_2C$ K720E | This paper | | Protein expression plasmid for *E. coli* (*Rattus Norvegicus* Munc13-1 (529–1735, °1408–1452)) K720E |
| recombinant DNA reagent | Pet28a-His6-Munc13-1-$C_1C_2BMUNC_2C$ R769E | This paper | | Protein expression plasmid for *E. coli* (*Rattus Norvegicus* Munc13-1 (529–1735, °1408–1452)) R769E |
| recombinant DNA reagent | Pet28a-His6-Munc13-1-$C_1C_2BMUNC_2C$ K603E, K720E | This paper | | Protein expression plasmid for *E. coli* (*Rattus Norvegicus* Munc13-1 (529–1735, °1408–1452)) K603E, K720E |
| recombinant DNA reagent | Pet28a-His6-Munc13-1-$C_1C_2BMUNC_2C$ K706E, R769E | This paper | | Protein expression plasmid for *E. coli* (*Rattus Norvegicus* Munc13-1 (529–1735, °1408–1452)) K706E, R769E |
| recombinant DNA reagent | Pet28a-His6-Munc13-1-$C_1C_2BMUNC_2C$ K603E, K706E, K720E, R769E | This paper | | Protein expression plasmid for *E. coli* (*Rattus Norvegicus* Munc13-1 (529–1735, °1408–1452)) K603E, K706E, K720E, R769E |
| recombinant DNA reagent | pLenti_f(syn)-NLS-GFP-P2A-Munc13-1-Flag | *Liu et al., 2016* | | Rescue construct protein expression plasmid for neurons (*Rattus Norvegicus* Munc13-1) |
| recombinant DNA reagent | pLenti_f(syn)-NLS-GFP-P2A-Munc13-1 K603E-Flag | This paper | | Rescue construct protein expression plasmid for neurons (*Rattus Norvegicus* Munc13-1 K603E) |
| recombinant DNA reagent | pLenti_f(syn)-NLS-GFP-P2A-Munc13-1 K706E-Flag | This paper | | Rescue construct protein expression plasmid for neurons (*Rattus Norvegicus* Munc13-1 K706E) |
| recombinant DNA reagent | pLenti_f(syn)-NLS-GFP-P2A-Munc13-1 K720E-Flag | This paper | | Rescue construct protein expression plasmid for neurons (*Rattus Norvegicus* Munc13-1 K720E) |
| recombinant DNA reagent | pLenti_f(syn)-NLS-GFP-P2A-Munc13-1 R769E-Flag | This paper | | Rescue construct protein expression plasmid for neurons (*Rattus Norvegicus* Munc13-1 R769E) |

*Appendix 1 Continued on next page*

*Appendix 1 Continued*

| Reagent type (species) or resource | Designation | Source or reference | Identifiers | Additional information |
|---|---|---|---|---|
| recombinant DNA reagent | pLenti_f(syn)-NLS-GFP-P2A-Munc13-1 K603E/K706E-Flag | This paper | | Rescue construct protein expression plasmid for neurons (*Rattus Norvegicus* Munc13-1 K603E/K706E) |
| recombinant DNA reagent | pLenti_f(syn)-NLS-GFP-P2A-Munc13-1 K720E/R769E-Flag | This paper | | Rescue construct protein expression plasmid for neurons (*Rattus Norvegicus* Munc13-1 K720E/R769E) |
| recombinant DNA reagent | pLenti_f(syn)-NLS-GFP-P2A-Munc13-1 K603E/K706E/ K720E/ R769E-Flag | This paper | | Rescue construct protein expression plasmid for neurons (*Rattus Norvegicus* Munc13-1 K603E/K706E/K720E/ R769E) |
| biological sample (*Unc13-a/b DKO*) | Primary neuronal cultures | *Varoqueaux et al., 2002* | | Freshly isolated from *Unc13-a/b DKO* mouse embryos |
| antibody | anti-VGLUT1 (Guinea pig polyclonal) | Synaptic System | 135,304 | 1:4,000 |
| antibody | anti-MAP2 (Chicken polyclonal) | Merck Millipore | AB5543 | 1:2,000 |
| antibody | anti-Munc13-1 (Rabbit polyclonal) | Synaptic System | 126,103 | 1:500 |

