## [Editor Report]

Munc13 is essential for exocytotic fusion of synaptic vesicles but the precise mechanism of action of this multidomain protein are not fully understood. The authors show that two different structural states of Munc13 are involved in, Ca^2+^-independent and Ca^2+^-dependent, synaptic vesicle priming. This is a significant contribution furthering our understanding of the complex multiprotein machinery responsible for the final steps in vesicle exocytosis. The study is comprehensive, careful, using a battery of different approaches that all substantiate the main conclusions of the work.

---

## [Decision Letter]

**Decision letter after peer review:**

Thank you for submitting your article "Control of neurotransmitter release and presynaptic plasticity by re-orientation of membrane-bound Munc13-1" for consideration by *eLife*. Your article has been reviewed by 3 peer reviewers, and the evaluation has been overseen by Reinhard Jahn as the Reviewing Editor and Olga Boudker as the Senior Editor. The reviewers have opted to remain anonymous.

Essential revisions:

All reviewers agree that this is an interesting and thorough study, conducted by two leading laboratories with complementary expertise, that deserves publication after appropriate revision. Moreover, the reviewers do not see the need for additional experiments in order to make the manuscript acceptable for publication. However, the reviewers feel that the interpretation of the findings should be more cautious and that some of the conclusions are overstated. Please check in particular the points raised by Ref. 1, most importantly the remarks concening the mechanistic interpretation (switch in orientation) of the mutant phenotypes. Mutations which reverse charges may, in addition to local effects, also have more global, electrostatic and thus less specific effects. Moreover, note that the MD simulation can only refine the proposed individual orientations but do not differentiate between alternative models. We also ask you to consider changing the order of the data presentation as suggested by Reviewer 1.

*Reviewer #1 (Recommendations for the authors):*

I strongly urge the authors to reverse the order of the manuscript so that the biochemical analysis of the mutants precedes the in vivo analysis. Make the mutants, characterize membrane binding, liposome tethering, and membrane fusion in vitro, then move on the much trickier business of trying to understand their consequences in vivo.

*Reviewer #2 (Recommendations for the authors):*

In Figure 7, the authors show that the calcium-independent phospholipid binding of the Munc13 mutants is significantly altered (Fig. 7A, upper panel). It is unclear why they examine calcium-dependent lipid binding at exceptionally high calcium concentrations (0.5 mM free calcium) (Fig. 7A middle panel), although calcium-dependent Munc13-mediated liposome fusion is already saturated at micromolar concentrations (Fig. 9B). Would it therefore not make more sense to delineate possible changes in calcium-dependent phospholipid binding of the Munc13 protein and its mutants at submicromolar concentrations?

*Reviewer #3 (Recommendations for the authors):*

It would be useful to have a methodological strategy to quantitate the ca^2+^-dependence of the transition from the TL to the TS state of the SNARE to further validate the model proposed by the authors. I understand, however, that this is technical challenge in the field and it is, of course, not required for this study. In any case, according to the model proposed by the authors, which changes in RRP size (specially in the amount of SNARE complexes in the TS state) would be expected from a mutant with a low release probability (e.g. a synaptotagmin mutant)? Perhaps this is a prediction that could be included in the discussion.

Page 25. A typo to be corrected: R603E is written instead K603E.

---

## [Author Response]

Essential revisions:All reviewers agree that this is an interesting and thorough study, conducted by two leading laboratories with complementary expertise, that deserves publication after appropriate revision. Moreover, the reviewers do not see the need for additional experiments in order to make the manuscript acceptable for publication. However, the reviewers feel that the interpretation of the findings should be more cautious and that some of the conclusions are overstated. Please check in particular the points raised by Ref. 1, most importantly the remarks concening the mechanistic interpretation (switch in orientation) of the mutant phenotypes. Mutations which reverse charges may, in addition to local effects, also have more global, electrostatic and thus less specific effects. Moreover, note that the MD simulation can only refine the proposed individual orientations but do not differentiate between alternative models. We also ask you to consider changing the order of the data presentation as suggested by Reviewer 1.

We very much appreciate the overall positive evaluation of the manuscript. We have followed the recommendations, toning down some conclusions, emphasizing the presence of two functional binding surfaces rather than two orientations, and switching the order of the data presentation.

Reviewer #1 (Recommendations for the authors):In addition to addressing/discussing the points within Public Review, I strongly urge the authors to reverse the order of the manuscript so that the biochemical analysis of the mutants precedes the in vivo analysis. Make the mutants, characterize membrane binding, liposome tethering, and membrane fusion in vitro, then move on the much trickier business of trying to understand their consequences in vivo.

We agree with this suggestion and have switched the order accordingly.

Reviewer #2 (Recommendations for the authors):In Figure 7, the authors show that the calcium-independent phospholipid binding of the Munc13 mutants is significantly altered (Fig. 7A, upper panel). It is unclear why they examine calcium-dependent lipid binding at exceptionally high calcium concentrations (0.5 mM free calcium) (Fig. 7A middle panel), although calcium-dependent Munc13-mediated liposome fusion is already saturated at micromolar concentrations (Fig. 9B). Would it therefore not make more sense to delineate possible changes in calcium-dependent phospholipid binding of the Munc13 protein and its mutants at submicromolar concentrations?

We used 0.5 mM Ca^2+^ to ensure saturation of the Ca^2+^-binding sites, as we do usually for our liposome fusion assays. We realize that millimolar concentrations of Ca2+ can induce lipsome clustering without the assistance of proteins, but these effects do not occur below 1 mM Ca^2+^ under our experimental conditions.

Reviewer #3 (Recommendations for the authors):It would be useful to have a methodological strategy to quantitate the ca^2+^-dependence of the transition from the TL to the TS state of the SNARE to further validate the model proposed by the authors. I understand, however, that this is technical challenge in the field and it is, of course, not required for this study. In any case, according to the model proposed by the authors, which changes in RRP size (specially in the amount of SNARE complexes in the TS state) would be expected from a mutant with a low release probability (e.g. a synaptotagmin mutant)? Perhaps this is a prediction that could be included in the discussion.

The reviewer raises a good point that is indeed difficult to address experimentally. Following the suggestion of the reviewer, we have slightly expanded the final paragraph of the discussion to briefly discuss potential roles of synaptagmin-1 and complexins, proteins that increase the release probability, on the RRP and the LS-TS equilibrium (pg. 27).

Page 25. A typo to be corrected: R603E is written instead K603E.

We have corrected the error.